# Automatic Detection of Focal Cortical Dysplasia Using MRI: A Systematic Review

**DOI:** 10.3390/s23167072

**Published:** 2023-08-10

**Authors:** David Jiménez-Murillo, Andrés Eduardo Castro-Ospina, Leonardo Duque-Muñoz, Juan David Martínez-Vargas, Jazmín Ximena Suárez-Revelo, Jorge Mario Vélez-Arango, Maria de la Iglesia-Vayá

**Affiliations:** 1Grupo de investigación Máquinas Inteligentes y Reconocimiento de Patrones, Instituto Tecnológico Metropolitano, Medellín 050013, Colombia; davidjimenez215489@correo.itm.edu.co (D.J.-M.); leonardoduque@itm.edu.co (L.D.-M.); 2GIDITIC, Universidad EAFIT, Medellín 050022, Colombia; jdmartinev@eafit.edu.co; 3Grupo de Investigación en Imágenes Médicas SURA, Ayudas Diagnósticas SURA, Carrera 48 # 26-50, Piso 2, Medellín 050021, Colombia; jxsuarez@sura.com.co (J.X.S.-R.); jmveleza@sura.com.co (J.M.V.-A.); 4Biomedical Imaging Unit FISABIO-CIPF, Foundation for the Promotion of the Research in Healthcare and Biomedicine (FISABIO), Avda. de Catalunya, 21, 46020 Valencia, Spain; delaiglesia_mar@gva.es; 5Centro de Investigación Biomédica en Red de Salud Mental (CIBERSAM-G23), 28029 Madrid, Spain

**Keywords:** deep learning, focal cortical dysplasia, image processing, machine learning, magnetic resonance imaging

## Abstract

Focal cortical dysplasia (FCD) is a congenital brain malformation that is closely associated with epilepsy. Early and accurate diagnosis is essential for effectively treating and managing FCD. Magnetic resonance imaging (MRI)—one of the most commonly used non-invasive neuroimaging methods for evaluating the structure of the brain—is often implemented along with automatic methods to diagnose FCD. In this review, we define three categories for FCD identification based on MRI: visual, semi-automatic, and fully automatic methods. By conducting a systematic review following the PRISMA statement, we identified 65 relevant papers that have contributed to our understanding of automatic FCD identification techniques. The results of this review present a comprehensive overview of the current state-of-the-art in the field of automatic FCD identification and highlight the progress made and challenges ahead in developing reliable, efficient methods for automatic FCD diagnosis using MRI images. Future developments in this area will most likely lead to the integration of these automatic identification tools into medical image-viewing software, providing neurologists and radiologists with enhanced diagnostic capabilities. Moreover, new MRI sequences and higher-field-strength scanners will offer improved resolution and anatomical detail for precise FCD characterization. This review summarizes the current state of automatic FCD identification, thereby contributing to a deeper understanding and the advancement of FCD diagnosis and management.

## 1. Introduction

Focal cortical dysplasia (FCD) refers to a group of developmental abnormalities in the cerebral cortex, which are characterized by localized architectural and cellular anomalies. It manifests itself as localized malformations in the cerebral cortex resulting from disrupted neuronal migration, abnormal cellular organization, and/or disordered cortical lamination. Histologically, FCD is characterized by disordered cortical layers, dysmorphic neurons, and white matter abnormalities [1]. This pathological condition has gained considerable attention in the medical field due to its close association with epilepsy. In fact, FCD is the most common cause of drug-resistant epilepsy in children and the second or third most common cause of anti-epileptic drug use in adults [2,3]. To effectively manage FCD, accurate diagnosis and appropriate treatment strategies are essential. Therefore, this review aims to provide a comprehensive overview of the automatic identification of FCD, focusing on recent advancements in this area.

Diagnosing FCD and determining the optimal approach to treat it poses significant challenges. Accurately identifying and precisely localizing FCD (depending on the clinician’s expertise) play a crucial role in guiding surgical resection, which has demonstrated considerable efficacy in achieving seizure control and enhancing the quality of life of many patients [4]. Recent advances in imaging techniques, particularly magnetic resonance imaging (MRI), have revolutionized FCD diagnosis, allowing the correct identification of patients who may benefit from resection surgery. The automatic processing of MRI images can achieve a detailed anatomical visualization of the brain, thus aiding in the identification of the subtle cortical malformations that define FCD [5,6,7]. Additionally, the integration of advanced imaging protocols—including high-resolution imaging sequences and specific MRI protocols tailored for FCD detection—has significantly improved the sensitivity and specificity of FCD diagnosis [5,8,9]. As a result, the use of techniques to automatically detect FCD has increased in recent years. In this review, we will primarily focus on automated FCD identification based on automatic computer-aided diagnostic tools supported by MRI, highlighting its strengths, limitations, and emerging techniques.

To the best of our knowledge, there is no review that focuses on automatic classification techniques of focal cortical dysplasia (FCD) in MRI images. However, there is a previous review that focuses on MRI post-processing techniques for the presurgical evaluation of patients [10]. This review was conducted in 2016 and focused on techniques to highlight lesions in the image, which can be achieved with the morphometric analysis program or sulcal morphology features. Moreover, it did not address recent advances in the field comprehensively, specifically the growing area of automatic FCD identification. Since then, multiple artificial intelligence techniques, such as machine learning and deep learning-based, have emerged and improved the detection of FCD.

In recent years, significant progress has been made in the development of automated techniques to aid in the detection and classification of FCD [11,12,13]. In this review, we aim to address this gap in the literature and present a comprehensive summary of current state-of-the-art approaches for automatic FCD detection. Our primary goal is to understand how FCD is detected from MRI images, with an emphasis on automatic or semi-automatic methods based on machine learning and deep learning techniques. To ensure a systematic and rigorous selection of relevant studies, we conducted a thorough literature search, applying the guidelines known as the preferred reporting items for systematic reviews and meta-analysis (PRISMA) [14].

This paper is organized into two main sections. The first section presents the materials and methods employed in the studies identified in our systematic review. We established a search equation to retrieve studies on the automatic detection of FCD in MRI. The search and identification of registers were performed in four databases. After a screening stage, 65 studies were included in this review, most of which were published from 2015 onward. We present some general characteristics of the datasets used in these studies, highlighting that there is no benchmark dataset for comparison in the field. This is because each research center has its own MRI dataset, and only six studies have a statement for data sharing. Furthermore, included studies were categorized according to the approach implemented in each study: visual, semi-automatic, or fully automatic methods. Then, we discuss the strengths and limitations of each approach, highlighting their potential contributions to accurate FCD identification. The second section draws conclusions from the literature previously reviewed, summarizing the current state of automatic identification of FCD and discussing potential directions for future research. By presenting a comprehensive synthesis of the literature in this rapidly evolving field, this review aims to contribute to a deeper understanding and the advancement of automatic FCD identification.

## 2. Materials and Methods

Figure 1 shows an example of an MRI image with FCD. The left panel displays one slice of the MRI on the axial view; the middle panel, the *ground truth* (i.e., the mask generated by a clinician); and the right panel, the segmentation of the FCD superimposed on the MRI image. Several techniques have been proposed for the detection and segmentation of FCD in MRI. These techniques can be grouped according to the type of algorithm they employ, the type of images or data they use, and the nature of the analysis (which can be mathematical, statistical, manual, or automatic, among others). In this review article, we propose a classification of these techniques (reported in the literature) into visual, semi-automatic, and automatic methods. The development of these techniques has led to a large amount of documentation, which requires using an already-established method to systematically compile and verify it in the form of a literature review. Moreover, the first review of this kind was only published in 2016 by Wang and Alexopoulos [10]. This article presents an up-to-date review of the scientific literature in this field and proposes a classification of articles based on the analysis model used by the authors. This review follows the guidelines of the PRISMA protocol for systematic reviews of scientific literature, which was initially focused on medicine but can be applied to other fields [14]. Figure 2 shows a flowchart of the steps followed in this study to carry out said protocol.

According to the PRISMA protocol, the articles to be reviewed must be selected by two people. In this review, they were chosen by the corresponding author and one of the co-authors. The first step was defining what research databases were going to be searched for articles. In this case, we selected Scopus, Science Direct, Web of Science, and PubMed to cover a large number of journals; we not only focused on technical aspects but also addressed the problem (FCD in MRI images) with a multidisciplinary approach, i.e., articles from medical and engineering fields. Then, the search string was formulated and tested several times, which resulted in the following string:
*(((FCD) OR (Focal Cortical Dysplasia)) AND ((Automatic) OR (Automated)) AND ((MRI) OR (Magnetic Resonance Images)))*

With the string above, the search was conducted in November 2022, and the search engines of the databases yielded 250 records of interest: 91 in PubMed, 66 in Scopus, 70 in Web of Science, and 23 in Science Direct. After a manual search, 110 articles were removed from the initial list of 250 because they were duplicates in different databases. After filtering out said duplicates, 140 articles remained. Among them, 70 articles were discarded because, after reading their title and abstract, it was found that they did not fit the research topic; another article was discarded because it was impossible to download. Finally, we were left with 65 papers that met the two following selection criteria: (1) their full text was read, and (2), based on their content, it was decided that they would be included in this literature review. Figure 3 shows the distribution of the selected papers by year of publication, suggesting that this is an open research area.

### 2.1. FCD Datasets

The studies identified with the PRISMA protocol and included in this review present different characteristics in their datasets. Table 1 shows the main characteristics of each dataset. We summarize the studies’ information regarding the following: The number of patients, the age range in years, the MRI sequence used, the scanner resolution, and whether the study has a statement of the dataset availability. One of the main problems that researchers face when conducting automatic classification studies of FCD is the generalization of their methodologies. As can be seen in this field, each research has its own data, and no studies have been conducted that allow comparison due to the lack of public data or benchmark datasets. In particular, only six studies have a statement that allows data sharing by request, but there is no public link to download the data.

### 2.2. General Framework

Several techniques have been proposed for the detection and segmentation of FCD in MRI. In this paper, we propose classifying these techniques into three categories: visual, semi-automatic, and automatic methods. The general framework that is followed by most of the researchers in the FCD identification is the following: (i) data acquisition, (ii) preprocessing, (iii) feature extraction, and (iv) classification of FCD. Figure 4 presents the general framework that is followed to conduct FCD detection.

*Data acquisition:* Most of the patients of the included studies in this review had refractory epilepsy and had undergone presurgical assessment with T1, T2, and/or FLAIR sequences. The magnetic fields used in the scanners were 1.5 T or 3 T. Only one retrieved study has a 7 T magnetic field.

*Preprocessing:* The objective of this step is to enhance and coregister the image for manual or automatic segmentation. The most common pre-processes include intensity correction, motion correction, removal of non-brain tissue, and segmentation of GM/WM. Additionally, most studies include an expert-based segmentation of the FCDs providing *ground truth* for comparison.

*Feature extraction:* This step is optional. Some studies only use the preprocessing step to enhance the image and perform manual segmentation. Others use computed features for training learning models. The following are the most common features: GM/WM intensity contrast, local cortical deformation (LGD), cortical thickness, mean curvature, sulcal depth, doughnut intensity, doughnut thickness, and WM hyperintensity on FLAIR [58].

*FCD classification:* After computing MRI features, the FCD classification is performed with visual, semi-automatic, or automatic methods. Visual methods involve extracting multiple features from MRI images. This can be done by looking for specific patterns of signal intensity or by combining MRI sequences with other imaging methods such as PET, to improve the contrast of the FCD in the image. Semi-automatic methods use specialized software to assist with the generation of features, such as composite maps [16], the identification of sulcus abnormalities [17], and diffusion parameters [19], among others. These features can improve the segmentation of FCD lesions. Automatic methods rely on the use of machine learning or deep learning algorithms for the detection and segmentation of FCD.

### 2.3. Visual Methods

Although early FCD studies were based on histological analyses and electrical activity obtained by electroencephalography (EEG), advances in medical imaging have shifted in favor of medical imaging. Nowadays, MRI is the most commonly used imaging technique for evaluating cortical dysplasia. This technique involves a complete study of the brain, in which coronal and transverse T2, coronal FLAIR, and coronal T1-weighted sequences are acquired. Some characteristics have been found to differentiate dysplasia in MRI images: cortical thickness, lack of definition of the junction between gray matter (GM) and white matter (WM), and abnormal patterns of sulci or gyri, among others [1].

More recently, other imaging techniques have been employed to perform presurgical segmentation. One of them is diffusion MRI, which is sensitive to axons rather than myelin. As a result, it can obtain a “more robust differentiation of cortex and white matter and improve malformations of cortical development delineation” [15] (p. 1). Another specialized and recently introduced MRI sequence for FCD delineation is 3D FLAIR. This technique showed “significant statistical difference (…) over conventional images” [70] (p. 1) in terms of showing differentiable features of FCD (such as cortical thickness).

### 2.4. Semi-Automatic Methods

Based on a previous study [7], Antel et al. sought to obtain higher contrast between lesional and non-lesional cortex. For that purpose, they employed image processing operators to model three of the most common attributes of FCD lesions on the T1-weighted sequence: increased cortical thickness, blurring of the GM-WM boundary, and hyperintensity of the signal compared to the normal cortex [16]. All the operators were voxel-based and produced three-dimensional feature maps that, in turn, were combined into a single composite feature map. Then, contrast, defined as the mean *z-score* of the lesional tissue, was calculated on both the original and composite maps for each patient. The percentage change in contrast from the original maps to the composite map was calculated. When evaluating the method on MRI images from 14 patients, it enhanced the contrast in 11 of them. Moreover, in one patient, the lesion was only visible with the help of the composite map. This result demonstrates that increasing the contrast between lesional and non-lesional tissue improves lesion detection by visual analysis of composite maps.

The main objective of a study by Roca et al. [17] was to perform a group-level quantitative analysis of abnormal sulcus patterns associated with FCD lesions in MRI-negative and MRI-positive patient images. Their study was based on (i) an automated quantitative analysis of sulcus abnormalities using a new sulcus descriptor called “sulcus energy”; (ii) three groups of subjects: MRI-positive, MRI-negative, and control subjects; and (iii) FCD lesions located in the central region. Their second aim was to evaluate the relevance of the *z-score* maps of sulcal energy in localizing the epileptogenic lesion at the individual level. A procedure consisting of four steps was used to characterize the sulcus–gyrus anatomy: (i) extraction of the cortical folds in the MRI T1-weighted sequence and their conversion to a graph-based representation of the cortex containing, for each fold, a list of morphological descriptors (area, depth, length, etc.) and a spatial organization relative to its neighbors; (ii) automatic recognition of the sulci using an algorithm based on a congregation of neural networks with optimized energy minimization by simulated annealing under the thesis that the final minimum energy is higher in patients with FCD; (iii) generation of the sulcus energy map calculated from the minimum final energy of the previous point; and (iv) calculation of the *z-score* maps of sulcus energy obtained from normalizing sulcus energy maps by dividing the difference between the patient’s sulcus energy and the mean of the sulcus energy of healthy subjects by the standard deviation of the controls. They used the MRI images of 29 right-handed patients. Among them, FCD lesions were histologically confirmed in 25 patients. The remaining 4 were MRI-positive patients. Additionally, they included 25 right-handed control subjects. At the group level, it was found that patients presented higher central sulcus energy than controls and that FCDs were associated with abnormal central sulcus patterns compared to healthy subjects. At the individual level, it was found that sulcus energy alone was insufficient to detect the FCD lesion. However, a correspondence was found between the maximum *z-score* and the location of the FCD lesion.

Semi-automated, software-based methods have also been used to identify FCD in MRI images as an adjunct to traditional methods, with positive results. In a study of 20 patients with histologically confirmed FCD, segmentations performed by 3 expert radiologists were compared to those produced by FreeSurfer software. The software segmentation detected two more cases than its manual counterpart (i.e., 25% more), demonstrating that the former can be used to detect FCDs that are invisible to conventional methods [18].

Using the latest advances in MRI software and hardware, Lorio et al. sought to determine whether diffusion parameters from multi-section models demonstrated consistent changes in probable FCD lesions [19]. To this end, they employed new diffusion models such as the *“Spherical Mean Technique”* (SMT) and *“Neurite Orientation Dispersion and Density Imaging”* (NODDI) to provide measures that can potentially produce more specific maps of abnormal tissue microstructure. They evaluated their method using the diffusion MRIs of 33 pediatric patients. NODDI maps were calculated for each brain voxel as three different sections: intra-neurite, extra-neurite, and CSF. SMT models were found by estimating microscopic diffusion tensors and multi-section microscopic diffusion maps. Lesion profiles were obtained for each diffusion map by averaging the values within the lesion mask and the homologous region along each sampling surface. Diffusion map profiles of lesions and homologous regions in all 33 patients were statistically compared to assess changes in FCD lesions. The maps obtained with these methods proved to have better contrast than FLAIR sequence images and, in some cases, had consistent signal changes specific to FCD, suggesting that they can improve imaging protocols in pediatric patients with epilepsy and be used as features in the automation of lesion detection.

In summary, studies included in this section were characterized by an automated component, usually based on image processing techniques to obtain features from MRI images via new descriptors, attributes, or diffusion maps to enhance the lesion visibility, and a human-guided component to perform the localization of the FCD.

### 2.5. Automatic Methods

#### 2.5.1. Mathematical Methods

Due to the deficiencies in segmentation of FCD in MRI that existing algorithms had at the time of their study and because they were primarily created for FCD detection only, Colliot et al. [20] proposed an algorithm based on level sets for segmenting FCDs in MRIs in the T1-weighted sequence. Their algorithm is supported by a 3D deformable model based on the level set method (the same 2D contour lines or 3D level surfaces). Instead of basing the model on gray image levels, the authors proposed a model guided by probability maps derived from visual features of FCD, such as cortical thickness, GM-WM boundary-blurring, and signal hyperintensity in the dysplastic region. Also, the starting point for the evolution of the level sets was taken from previous classification work by the same authors. The region competition method forms the lesion region to segment the image into several regions by moving the interfaces between them. The evolution of the interfaces is given by functions derived from the FCD features, which indicate the membership to each region. Thus, region competition occurs at each point between the lesion class and the most likely non-injury classes: GM, WM, and cerebrospinal fluid (CSF). The study employed a database of 24 MRI-positive patients. The classification algorithm to initiate the level set detected 18/24 (75%) lesions; therefore, the method was only applied to those cases. The automatic segmentations were compared to their manual counterparts using a similarity index, and the automatic level sets exhibited a 62% similarity to the manual segmentations.

In a subsequent study, Colliot et al. [21] used the same methods as in [20], but this time supported by two successive 3D deformable models. The first model, as in their previous work, was given by the probability maps obtained from visual features of FCD on T1-weighted MRI: cortical thickness, relative intensity, and gradient. The second model took the previous model’s output to expand the lesion, by gradient vector fluxes, from the GM-WM boundary to other underlying or overlying cortical regions. The algorithm was evaluated on the same 18 patients as in the previous study and obtained a similarity of 73% to the images manually segmented by an expert.

Despotovic et al. proposed an improved graph cuts algorithm for more accurate segmentation of the cerebral cortex by integrating Markov random fields based on an energy function [22]. Their method was based on segmenting the cerebral cortex on T1-weighted MRI using three labels: GM, WM, and CSF. The energy minimization problem was solved by calculating the minimum cost in a graph equal to the total energy of the corresponding segmentation. The method was evaluated with simulated images and MRI of eight patients using the Dice coefficient. In that study, a 95% similarity was obtained. However, in the paper, it is unclear if this is an average of the eight patients.

Snyder et al. implemented a new method to describe and detect FCD lesions [23]. First, they created a model based on a rotationally invariant and multi-contrast 3D local feature implementation that describes the normal variability of the cortex in healthy subjects. With the features of this model, a latent representation of the data was created to enable the direct detection of outlier cases in the multivariate feature space. In patients, many FCD lesions appear as outlier cases, but with underlying features similar to usually atypical regions. Consequently, the similarity between the features underlying the FCDs is assessed. The automatic detection of FCDs was based on these similarity maps, in which the mean feature vector for each cortical patch is projected onto the average unit vector of FCDs. With a database of 15 patients and 30 control subjects, this study was able to automatically detect 12/15 lesions with an optimal set of hyperparameters for a sensitivity of 80% and an area under the curve of 0.96. Among the 12 detected lesions, 11 belonged to MRI-positive patients and only 1 to MRI-negative patients.

Lotan et al. [24] researched the neocortical laminar architecture in patients with FCD and periventricular nodular heterotopia (PNH) to detect cortical abnormalities. They compared FCD and PNH patients with healthy subjects employing logistic regression. For this purpose, they calculated cortical layers in T1-weighted images by finding the GM probability map. Using the GM map, a histogram of GM values in the T1-weighted image (*T1 corticogram*) was created and used to classify the T1 layers. The *T1 corticogram* was then fitted to a t-distribution model, which resulted in six t-distributions that presumably account for the distributions of the six cortical layers. Using Bayes’ rule, six cortical probability maps were calculated, one for each T1 layer demonstrating a laminar pattern. To determine the laminar composition of the different cortical areas, an atlas of the 78 cortical regions of interest was used. Then, these regions were divided into 1000 sub-areas of equal volume. The most significant finding in that study was the observation of the extent of abnormal cortical architecture in patients compared to control subjects—and that this architecture extends beyond the presumed epileptogenic areas as seen in conventional MRI analysis.

In this section, we gathered methods that perform mathematical analysis on MRI images to detect FCD lesions. This process involves the solution of mathematical expressions, such as differential equations and Markov fields, or building deformable models and probability (*z-score*) maps based on visual features of FCD. These approaches allow the authors to highlight FCD lesions by extracting brain tissues from a patient MRI image and comparing them to the average map of healthy subjects. Unlike semi-automatic methods, the studies of this section do not require the human-guided component.

#### 2.5.2. Automatic Methods Based on Volumetric
Morphometry

The first automatic methods for FCD detection and segmentation were based on computer-aided calculations of different mathematical and statistical models to characterize brain structures, create maps of healthy brains, and make comparisons with brain maps of FCD patients. These models were implemented for FCD feature extraction in MRI; later, they were widely used to train machine learning algorithms.

##### Voxel-Based Morphometry (VBM)

In this literature review, the first study that implemented VBM to fully automate the detection of FCD was authored by Kassubek et al. [25]. They calculated GM density maps, where each voxel encoded the GM concentration and its corresponding position in the original MRI. The GM density maps of 30 healthy subjects were first calculated, obtaining a single overall map by calculating the mean of all of them; then, the maps of seven patients with FCD were calculated. To find the lesions in the GM density maps of those seven patients, the mean GM density map of the healthy subjects was subtracted at the voxel level to calculate the global and local maxima of the resulting difference in the images. These maxima were compared to the standard deviation of the density map values of healthy subjects at their corresponding locations. With this method, lesions were found in 6/7 patients, where the global maximum (i.e., the maximum difference in GM density between patients and healthy subjects) corresponded to the location of the lesion as detected by conventional visual analysis.

As an extension of the work in [25], where FCD lesions were made more visible using gray matter density maps, Huppertz et al. [6] proposed a new method based on VBM. The said method includes a novelty, i.e., different FCD features (such as GM-WM junction blurring) are rendered more visible to improve the delineation and recognition of FCD. For each patient (taken from a database of 25 patients with histologically proven FCD), two three-dimensional feature maps derived from the original T1 sequence were calculated: (1) a “junction image” that highlights brain regions where there is blurring at the GM-WM junction and (2) an “extension image”, which is the same gray matter density map that was calculated in [25]. Likewise, these two maps were calculated for each one of the 53 control subjects, and, by finding the mean of all 53 maps, two maps of healthy subjects were generated. As in the previous work, Huppertz et al. expected that the maximum difference between the “junction image” and the original T1 would improve lesion visualization, i.e., increase the contrast between normal and FCD brain regions. The method found lesions in 18/25 patients by applying the “junction image” map, showing that blurring at the GM-WM boundary is characteristic of these lesions. Combining the two (the method employing the gray matter density maps in the previous paper and the method based on blurring at the GM-WM boundary in their paper), 21/25 lesions were detected, 4 of which were not recognized in conventional MRI analysis.

Aiming to further improve the visualization of blurring at the GM-WM boundary to automate the detection of FCD in patients whose MRI shows this feature and faced with the number of False Positives (FPs) introduced by gradient-based methods (especially in regions where the intensity of the GM-WM transition is weak)—Qu et al. [26] proposed a new algorithm they called “Iteration of Local Searches in the Neighborhood” (ILSN). This new method measured the width of the blurred region rather than the strength of the gradient. Their study included eight T1-weighted sequence MRI images from eight patients whose FCD manifested on the image as blurring at the GM-WM boundary. After preprocessing the MRIs (brain extraction, bias correction, and interpolation), the authors estimated the GM-WM boundary region by extracting the GM and WM maps using the FSL-FAST method. Then, they calculated the boundary region as those voxels that have both GM and WM information. Then, the potential map was modeled analogously to an electric field, where the field strength at a given point, i.e., the value of the voxel, equals the negative of the potential gradient at that point. The results of the evaluation of this method indicated that it had a better ability to identify FCDs characterized by GM-WM boundary blurring than gradient-based methods.

Colliot et al. [21] used VBM to investigate changes in the GM of individual patients. Previously, many of these authors [7,16] had noted that dysplasia manifesting in hyperintensities in GM in the T1-weighted sequence are usually ruled out when they are labeled as part of WM. Their 2006 study, which included 27 patients and 39 controls, compared the GM maps of each subject with the average GM map of all the control subjects, yielding a *z-score* map for each individual. The resulting *z-score* maps showed that in 21/27 subjects the lesion coincided with an increase in GM-WM intensity.

Feature maps highlighting the WM-GM boundary blurring feature for automatic FCD detection depend on small differences in signal intensity, and T2 sequences are highly sensitive to subtle differences in signal intensity. To address these issues, House et al. [27] qualitatively and quantitatively evaluated the performance results of morphometric analysis for FCD detection in *z-score* maps. The latter were obtained by comparing the GM-WM boundary maps in T1 and T2 sequences of 20 patients with FCD against the average maps of control subjects. The results of the qualitative analysis showed that, in 16 out of the 20 patients, the blurring feature at the GM-WM border was visualized with higher contrast in the T2 sequence. In three patients, there was no substantial difference between the two sequences, and, only in one case, the T1-weighted sequence showed better contrast. For the quantitative evaluation, the authors compared the mean *z-score* inside and outside each patient’s FCD for the T1 and T2 sequences. In 19 cases, the ratio was higher for the T2 sequence than for its T1 counterpart, confirming the result of the qualitative comparison. The remaining case, where the T1-weighted sequence showed better contrast in the quantitative evaluation, corresponds to the same case in the qualitative evaluation. It can be concluded from this study that the T2 sequence is more appropriate to evaluate or detect FCD when it manifests itself as blurring at the GM-WM boundary.

Since gradient magnitude maps used to model the GM-WM boundary blur cannot quantify the width of the GM-WM junction, Qu et al. [28] proposed a new algorithm called “Local Directional Probability Optimization” (LDPO), which aims to detect and quantify the width of the GM-WM boundary within lesional areas. First, the GM-WM boundary is extracted from the MRI employing a Markov random field algorithm. Then, the local directions of the voxels on the GM-WM boundary passing between the GM and WM are generated by considering the GM-WM boundary as an electric potential field. Finally, the optimal local directions are estimated (by iterative search in a neighborhood) to find the width of the GM-WM boundary. The authors evaluated the effectiveness of the proposed method on brain MRIs of 10 patients and 31 healthy subjects. The results indicate that the width of the GM-WM boundary blur in the lesional region, calculated by the LDPO method, is larger than in the non-lesional regions. The proposed method obtained higher *F-score* values employing the GM-WM blur within the lesional region than using feature maps.

A variant of VBM—which the original authors [71] named *MorphoBox* and is based on another type of biomarker known as “Volume-Based Morphometry” (VolBM)—was used by Chen et al. [29] to estimate whether volumetric abnormalities (atrophy, hypertrophy) can aid in the automatic detection of FCD. Unlike VBM, *MorphoBox* provides volumes of different brain structures of interest and compares them to those of healthy subjects from a similar population in terms of age and gender. This technique focuses on a single region of interest composed of multiple voxels, where it is assumed that the cumulative effect of abnormalities in each voxel becomes more apparent than when each voxel is considered individually (as is the case of VBM), thus increasing its sensitivity. Also, *MorphoBox* considers both GM and WM, which may improve the detection rate of FCD, as abnormalities may not be restricted to a single tissue. Evaluated in 16 patients with histologically confirmed FCD, this technique found abnormalities (such as atrophies and hypertrophies) in all of them. Epileptogenic zones in brain structures with abnormal volumes were found in 87.5% of their patients (14/16). Among them, 71.4% (10/14) presented regions with atrophic abnormality. These findings suggest that FCD lesions are more likely to be found in regions with atrophic volumes than in those with hypertrophic volumes.

Qu et al. [30] fused multiple classifiers (optimizing their parameters) with a genetic algorithm to take advantage of their high sensitivity and reduce the number of false positive (FP) results. To do so, they first found the maps of 6 features in 10 patients with FCD: cortical thickness, gradient, relative intensity, GM-WM border, curvature, and furrow depth. The classifiers selected by the authors were the Naïve Bayes, linear discriminant analysis (LDA), Mahalanobis discriminant analysis (MDA), and quadratic discriminant analysis (QDA). Each feature map was compared to the average map of 31 healthy subjects to find the corresponding *z-score* map. Then, each *z-score* map was passed through each classifier, and an *F-score* was calculated for each classifier. Finally, the average *F-score* per map was estimated to select the *P* best maps. The experimental results of that study showed that their method could detect FCD in 9 out of 10 patients and correctly classify all 31 control subjects with a specificity of 100%.

Feng et al. [31] proposed a method to detect FCD automatically, in negative FLAIR sequences, based on the measurement of cortical thickness and its standard deviation. The authors first extracted the average cortical thickness map and its standard deviation from a set of T1-sequence MRI images of 32 healthy subjects. Then, for FLAIR sequences from a database of six patients with histologically confirmed FCD, a *z-score* map was obtained by subtracting the average cortical thickness map from the cortical thickness map of each patient and dividing it by the standard deviation of the average cortical thickness of the healthy patients. This method detected 3/4 lesions located in non-temporal areas, but it failed to detect lesions in the temporal lobe of three patients.

##### Statistical Parametric Mapping (SPM)

SPM, a popular neuroimaging analysis method developed by Friston et al. [72], implements a VBM-based methodology. As indicated by [73], the implementation of SPM involves the extraction of multiple probability maps of brain tissues (such as GM and WM) by applying the Bayesian segmentation algorithm, smoothing by the Gaussian kernel, and normalization. Finally, based on the concentration of GM effects in the case under study, a statistical analysis is performed using the “General Linear Model” (GLM).

The methodology presented by Srivastava et al. [32] takes advantage of the intensity features of FCD lesions in the T1-weighted sequence and has a novelty: a statistical analysis of the FCD feature maps using SPM and GLM. From the GM and WM maps, a lesion-specific feature map is constructed by calculating the ratio between cortical thickness and absolute gradient intensity in the MRI for each GM voxel. The feature maps in a database of 17 patients were contrasted with similar maps obtained from a control group of 64 healthy subjects. The model characterized FCD lesions as cortical areas, showing thickening in the cortex and blurring at the GM-WM boundary. Their method detected and correctly localized the FCD lesion in 9/17 (53%) patients using a threshold that minimizes FP and 12/17 (71%) lesions using a threshold that allows a more significant number of FP results.

Focke et al. [33] performed an SPM analysis on the T2-FLAIR MRIs of 25 patients and 25 control subjects. The methodology consisted in taking T1-weighted sequence MRIs and registering them in a T2-FLAIR MRI space, which was followed by tissue segmentation in the T1-weighted sequence. Then, the FLAIR images were normalized in terms of intensity according to the “robust” value of mean WM intensity, and the skull was extracted using the Brain Extraction Tool (BET). Finally, convolution was performed between the FLAIR images without a skull using a Gaussian kernel and an SPM GLM to compare each patient against the control group. With this algorithm, lesions were found in 22/25 (88%) patients, with only one false positive in one of the control subjects.

The following year, they used the same SPM algorithm as in the previous article for the automatic detection of FCD [33], but this time with two new features [34]. First, the study was expanded to a database of 70 patients that included MRI in the T2-FLAIR sequence—although with the same number of control subjects (25). Second, unlike in [33] (where the FCDs were reported by two expert radiologists), in their 2009 paper, the authors had to rely on video-EEG telemetry as it was not possible for the experts to detect the FCDs. The performance of the algorithm was lower in this case, finding only 10/70 lesions in the patients (14.3%). The authors attributed this finding to diverse causes. For instance, during the video-EEG telemetry, some patients did not present seizures; therefore, their FCD was estimated based on interictal findings and clinical history.

Aiming to determine the sensitivity and specificity of VBM analysis and compare its results with other methods, Wong-Kisiel et al. [35] used the SPM-12 method to compare the morphometric characteristics of MRIs from a cohort of 39 patients with histologically confirmed FCD (including 20 adult and 19 pediatric patients) to those of a control group (consisting of 64 adults and 41 pediatric patients)—all in the T1-weighted sequence. After extracting the GM, WM, and CSF masks, the spread image was created by subtracting the mean GM of the control subjects from the GM mask of each patient and then dividing it by the standard deviation of the GM mask of the controls, which yielded a *z-score* map. The junction image was created similarly but, instead of the GM mask, the GM-WM border mask was used, and the *z-score* map was obtained. This method detected FCD lesions in 36/39 cases (92%). Among those 36 cases detected, the extension image found 34 lesions; and the junction image, 29.

Lin et al. [36] used a quantitative analysis of positron emission tomography (PET) images in combination with SPM techniques to identify abnormalities present in the images of a cohort of MRI-negative patients. They aimed to test the following hypothesis: the regions co-identified in the SPM and PET analyses are epileptogenic. T1-weighted sequence MRIs of 104 patients were processed by the SPM-12 methodology. The junction image of each patient and its statistical difference with respect to a control patient database were calculated to establish the *z-score* map and find the FCD in the form of blurring at the GM-WM boundary. The PET images were quantitatively evaluated to generate QPET statistical maps. Only those images where FCD was detected by SPM-12 analysis were contrasted with their respective QPET image. A region was labeled positive if and only if, in both cases, it matched; otherwise, it was labeled negative. With the SPM-12 analysis, 82/104 cases were detected, out of which 77 were corroborated by QPET analysis. The sensitivity was 74% with a QPET threshold SD=−1 selected according to the standard deviation concerning control subjects.

A study based on SPM-12 with pediatric patients was proposed by Wang et al. [37]. They included 78 MRIs (40 from 3T scans and 38 from 1.5T scans) in the T1-weighted sequence of MRI-negative patients and 370 control subjects, in both cases aged between 5 and 21. Following the typical steps in MRI morphometric analysis by SPM, malformations were found in 44 of the 78 patients (56%), 7 of them with lesions in multiple regions. Among the 44 patients whose MRI was positive, 21/40 (52%) corresponded to samples of 3T MRI, and 23/38 (60%) cases were found in 1.5T scans. The authors did not interpret this as a significant statistical difference between the two groups to suggest that one type of scanner is more suitable for automatic FCD detection. Table 2 summarizes the results obtained in studies that have implemented SPM.

Voxel-based morphometry and statistical parametric mapping are widely used methods for detecting FCD lesions. These methods seek to characterize brain structures and identify specific visual features of FCD, such as cortical thickness, GM-WM boundary-blurring, curvature, and signal hyperintensity, among others. These are considered standard features for FCD lesions detection and are leveraged in the studies included in this section to improve lesion detection accuracy.

#### 2.5.3. Automatic Methods Based on Machine Learning

This section focuses on traditional machine learning methods that have been shown to perform well in image analysis and, therefore, have been tested in various studies to measure their performance in the automatic detection of FCD lesions. This literature review revealed that Naïve Bayes classifiers and support vector machines (SVMs) are the most commonly employed algorithms in this domain. Additionally, decision trees and XGBoost have also been explored, although to a lesser extent.

##### Bayes Classifier

Perhaps the earliest attempt to solve the problem of classifying voxels as FCD or non-FCD in MRI using machine learning techniques was made by Antel et al. [38]. They used a two-stage Bayes classifier trained on images from a cohort of 18 patients with histologically confirmed FCD and 14 control subjects. In the first stage, the classifier was trained using first-order morphometric features extracted from the image after segmenting it into GM, WM, and CSF. Then, computational models of cortical thickness, blurring of the GM-WM boundary, and hyperintensity in the T1 signal were created from these maps of brain structures. The purpose of this first stage was to classify the areas as lesional or non-lesional. In the second stage, the classifier was trained using three second-order texture features selected by the authors: angular second momentum, difference entropy, and contrast. This stage was employed to reclassify those areas classified as lesional in the first stage. The combined classifier identified 17/20 manually labeled lesions.

For their part, Yang et al. [39] used the Naïve Bayes algorithm to classify 21 subjects with FCD. In contrast to [38], the authors did not segment the image into brain structures but selected features that were not dependent on such segmentation. In addition, they focused on classifying neighborhoods of voxels rather than individual voxels. The computed features were cortical thickness, the skewness of cortical thickness, the kurtosis of cortical thickness, absolute gradient, and the variance/skewness/kurtosis of the gradient vectors’ orientations. This approach enabled them to identify lesions in 21/21 patients.

In a subsequent study, Yang et al. [40] used the same Bayes classifier described in [39], trained with the same features, as the input stage for a second classifier. This new classifier, an SVM with a third-order polynomial kernel, was employed to reduce the number of FPs. In this methodology, the groups marked as lesional by the first classifier had the same features computed. Then, the differences with respect to the first classifier were also computed and used as input for the second classifier. Using this algorithm, the authors achieved a detection rate of 88%.

Strumia et al. [41] trained a Bayes classifier using one intensity feature, two texture features, and two shape features extracted from T1-weighted and FLAIR MRIs. The intensity feature was extracted from both T1-weighted and FLAIR sequences by creating a 2D distribution to highlight the hypointensity of the T1 signal and the hyperintensity of the FLAIR signal. The first texture feature accounted for the magnitude of the gradient of the T1-weighted sequence, which captured the blurring of the GM-WM boundary. The second texture feature highlighted the low definition at the GM-WM boundary in FCD lesions and was expressed as the variance in the spatial orientation of the gradient calculated in the T1-weighted image. Lastly, the two shape features were extracted from the T1-weighted sequence, visualizing FCD as either an amorphous mass of brain tissue or a region where GM and WM tissues merged without a discernible boundary. The first feature, fractional anisotropy, was defined as the eigenvalues of the Hessian matrix in a 2 ${\mathrm{mm}}^{3}$ window. The second feature measured the asymmetry of the local cortical thickness histogram. These features were computed for all MRIs in a database of 11 patients and 20 control subjects. The features computed for the patients and the control subjects were compared at the voxel level. Assuming a Gaussian distribution for the probability of a feature $f_{i}$ belonging to normal tissue, the likelihood that it belonged to an FCD lesion was determined as the complement. The features were assumed to be statistically independent, and classification was performed using a Naïve Bayes classifier, resulting in a spatial probability map of FCD. The average results for all patients were an accuracy of 51%, a sensitivity of 15%, and a Dice coefficient of 13% with respect to the real labels (*ground truth*).

Kulaseharan et al. [42] implemented a two-stage Bayes classifier based on the one described in [38] but applied to T1-weighted, T2-weighted, and FLAIR MRIs of 54 pediatric patients and 13 control subjects. This classifier was initially trained on morphometric features only. Subsequently, the voxels classified as lesions were reclassified using texture features. The sensitivity of this method was 97% in MRI-positive patients and 70% in MRI-negative patients.

Lastly, the most recent study using a Bayes classifier was conducted by Feng et al. [43]. They employed features extracted from FLAIR MRIs of seven MRI-negative patients to train a three-stage Naïve Bayes classifier and generate a 3D lesion map. This classifier yielded a sensitivity of 87.5%. Table 3 summarizes the results obtained by different authors using the Bayes algorithm.

##### Support Vector Machines (SVMs)

SVMs are machine learning algorithms widely used for the automatic detection of FCD lesions. In [40], SVMs were employed in the second stage to reclassify voxels previously classified as non-lesional by a Bayes classifier. This methodology yielded a sensitivity of 88% in a database of images from 21 adult patients.

Hong et al. [44] used an SVM to classify 41 subjects with histologically confirmed FCD (13 type I; 28 type II) and 41 control subjects. To this end, the authors performed an individual- and group-level analysis of cortical thickness and folding complexity (curvature maps). The SVM was able to predict histologic FCD subtypes with an average accuracy of 98% (100% in FCD type I; 96% in FCD type II). In the lateralization of the epileptogenic focus, the accuracy was 92% in FCD type I and 82% in FCD type II. Moreover, the SVM predicted surgical outcomes with an accuracy of 92% and 82% for patients with FCD type I and type II, respectively.

El Azami et al. [45] implemented a special version of the SVM algorithm, known as one-class SVM (OC-SVM). The OC-SVM was trained with negative (normal) examples from a database of 77 healthy subjects to detect voxel-level abnormalities in a group of 11 patients with 13 lesions (3 MRI-positive lesions). In addition, two feature maps were computed for all the subjects to train the algorithm: (1) the extension map that showed the extension of the GM into the WM and (2) the junction map that depicted the junction between the GM and the WM. Moreover, mean and standard deviation maps of both feature types were computed based on the control subjects. Finally, a *z-score* map was obtained by subtracting the mean map from the value of the individual map and dividing it by the standard deviation map. At the voxel level, the OC-SVM was trained with each voxel being associated with an OC-SVM classifier. The input features were represented by the voxel values of the corresponding coordinates in the extension and junction maps. Remarkably, this algorithm was able to detect 100% of the lesions in the MRI-positive cases (3/13). For the MRI-negative cases (10/13), it detected 7/10 lesions, for a 70% detection rate.

Other authors [46] focused on automatically detecting subtle or visually unidentifiable FCDs by combining features extracted from MRI and PET images of a cohort of 28 subjects with histologically confirmed FCD. Morphological features, such as cortical thickness and sulcus depth, as well as intensity-based features (including GM relative intensity and gradient at the GM-WM interface) were extracted from the images. The classification process involved two stages. In the first stage, the SVM classifier identified lesional vertices with the highest detection rate. The lesions detected were the input for the second-stage patch-based classifier, which, through statistical analysis of maps, aimed to remove FPs generated by the first classifier. The experimental analysis demonstrated that including both MRI and PET images using the two classification stages yielded the maximum sensitivity (93%).

Finally, Wang et al. [8] proposed a comparative analysis of machine learning algorithms, specifically Gaussian processes for machine learning (GPML) and SVM. This study was found to be the only one that has used a GPML algorithm and also the first to incorporate diffusion tensor imaging (DTI) training alongside T2-weighted MRI images from 12 patients with radiologically defined FCD. The comparison between the two algorithms revealed that the GPML-based classifiers outperformed their SVM counterpart. Using the area under the curve (AUC) metric, it was found that the mean AUC of the GPML-based classifier was AUC=0.76; and that of the SVM classifier, AUC=0.69.

Table 4 shows the results obtained in the different studies that have used SVMs. One study [8] was excluded because it employed a different evaluation metric that prevents a direct comparison. Based on this table, it can be inferred that better performance was achieved in the second study, which can be attributed to the use of a larger number of training images. Furthermore, the performance of the classifiers has improved as more images have been available for training.

##### Decision Trees

Decision trees have emerged as another approach for automating FCD detection. Gill et al. [47] employed surface-based morphological and intensity features extracted from T1-weighted, T2-weighted, and FLAIR MRIs together with a FLAIR/T1 ratio map to train a two-stage classifier. Their approach was validated with 41 patients with histologically confirmed FCD and 38 control subjects. In the first stage, vertex classification was performed to maximize sensitivity. To this end, the RUSBoost classifier, which combines random undersampling (RUS) and reinforcement learning (AdaBoost), was used to address the class imbalance by removing samples from the class with the highest number of examples. The output of the first classifier was clusters of lesional voxels that were then used as input for the second classification stage, aimed at improving the specificity of the model by removing FPs predicted in the first stage. The model achieved a sensitivity of 83% with 34/41 lesions detected and a specificity of 92% with no lesions found in 35/38 healthy controls.

Lin et al. [48] developed a classifier trained with surface-based cortical features for accurately localizing FCD type IIb lesions on images of 22 patients with histopathologically confirmed FCD. By integrating T1-weighted and FLAIR MRI and PET images, the authors extracted the following features: cortical thickness, FLAIR intensity, sulcus depth, LGI, GM area ($\mathrm{mm}{}^{2}$), GM volume ($\mathrm{mm}{}^{3}$), PET intensity, and GM-WM contrast intensity. The XGBoost classifier selected by the authors was trained with MRI alone, PET alone, and a combination of PET and MRI. The best performance was achieved by the classifier trained with the combination of PET and MRI features, which yielded a sensitivity of 93%, a specificity of 91%, and an FP rate of 9%. Table 5 shows the results obtained in the different studies that have used Decision trees.

##### Other Machine Learning Methods

To automate the detection of FCD in MRI-negative patients with extratemporal epilepsy, Hong et al. [49] used the Fisher linear discriminant analysis (LDA), which separates classes based on the linear combination found in the input data. The classifier was trained using surface-based morphological (cortical thickness, sulcus depth, and curvature) and intensity-based (relative intensity and gradient) features and tested on a database of 19 patients, 24 healthy controls, and 11 controls with temporal lobe epilepsy. The algorithm detected lesions at the exact location of manual labels in 14/19 patients, yielding a sensitivity of 74% and a specificity of 100% (no lesions were found on the MRIs of control subjects). In addition, the classifier was evaluated on a second database of 14 patients and 20 control subjects, yielding a sensitivity of 71% and a specificity of 95%.

Ahmed et al. [9] implemented a set of logistic regression classifiers to classify lesional and non-lesional vertices. The algorithm was trained with morphometric features (cortical thickness, GM-WM contrast, sulcus depth, curvature, and a measure of global brain deformation based on surface properties of gyri and sulci) of MRI-negative patients with histologically verified FCD. The authors employed an innovative strategy in which separate classifiers were trained on abnormally thick and abnormally thin regions. Additionally, different classifiers were trained for the gyral wall, sulcus, and crown to optimize the detection of lesions present in the sulcal bottom. This method detected lesions in 6/7 MRI-positive patients (86% detection rate) and 14/24 MRI-negative patients (58% detection rate).

Seeking to reduce the number of FPs predicted by classification algorithms, Qu et al. [50] implemented a unanimous voting of the multiple classifiers (UVMCs) method built upon a combination of different classifiers, including Naïve Bayes, LDA, MDA, and QDA. Each of these classifiers, individually, was a component of the UVMC algorithm and was fed with features extracted from MRIs of 10 FCD patients and 31 control subjects. The purpose of this method was to predict whether voxels belonged to the FCD-positive or FCD-negative class. According to the authors, when all classifiers labeled a voxel as FCD-positive, the UVMC method reclassified it as FCD-positive; otherwise, it reclassified it as FCD-negative, thus reducing the number of FPs. Although the number of true negatives was also reduced, most of the FCD-positive region was preserved. While the authors did not specify the numerical values of the experiment, the evaluation of the model showed that the UVMC method reduced the number of FPs compared to other algorithms described in the literature, such as the two-stage Bayes classifier proposed in [38] and each of the classifiers that composed the UVMC separately.

In a subsequent paper, Qu et al. [51] proposed a similar UVMC method using the same classification algorithms and some new features to distinguish between lesional and non-lesional regions. The first novelty in that study was that, after extracting the features from the images, different feature sets were formed to train each of the classifiers (Naïve Bayes, LDA, MDA, and QDA) to select the best feature set according to the performance of the classifiers. After selecting the best feature set, the voxels were classified as lesional or non-lesional using the UVMC algorithm, as in the previous study. Another novelty in the new algorithm was that the classification was refined through a connected region analysis by binarizing the predicted image, applying a mask to remove noise from regions marked as lesional, and then removing those regions whose size was smaller than a threshold. This study also included T1-weighted MRIs of 10 FCD patients and 31 control subjects. A total of 8/10 participants were correctly identified as patients, while 30/31 control subjects were classified as healthy.

Lee et al. [52] proposed the only unsupervised learning algorithm found in the literature. This is an ambitious study in which *consensus clustering* (a method that combines the results of multiple clustering algorithms) was applied to structural features of FCD on MRIs to identify the groups or classes that make up a lesion. In addition, the relationship between the FCD classes and their histopathology was evaluated. The database employed in the study consisted of 46 patients with FCD and a control group of 35 subjects. To group lesion regions, surface-based features were extracted from the entire MRI volume: cortical thickness, normalized FLAIR intensity, gradient, T1/FLAIR ratio, and functional derivatives in fMRI. An analysis of these features was then performed on the mask of the region labeled as lesional by an expert. The consensus clustering algorithm was applied to the features in this region to estimate the stability matrix, which stored the probability that the intralesional FCD features belonged to the same group or class. Finally, the authors trained a two-stage *XGBoost* classification algorithm to assign vertices labeled as lesional to a specific FCD class, using the same features as the clustering algorithm along with the values of their means in neighboring vertices. The first stage of the classifier sought to maximize sensitivity and consisted of four classifiers, each tuned to one of the four classes discovered by the clustering algorithm, whose output fed a meta-classifier responsible for making the final prediction. The purpose of the second stage was to remove the FPs detected in the first stage, using the same four classifiers followed by a meta-classifier. As a result, the clustering algorithm grouped the FCD type II lesions into four classes with different structural, histopathological, and functional impact profiles. Regarding FCD detection, the classifier trained with lesional class information detected more lesions than the classifier trained without said information (77 ± 3% vs. 73 ± 3%).

This section included machine learning-based methods with demonstrated performance in image processing. These were mainly supervised learning techniques involving extracting image features such as cortical thickness and blurring of the GM-WM boundary, among others. Some of the most frequent algorithms that appeared in this section were SVMs, Naïve Bayes, boosting, and decision trees. To a lesser extent, we also found some unsupervised clustering techniques with lower performance than that achieved by the SVM and boosting algorithms.

#### 2.5.4. Automatic Methods Based on Deep Learning Algorithms

VBM methods—although practical and robust in their implementation—have some inherent limitations. As explained in [53], they do not contain a spatial relationship across the cortical surface, so any registration errors may result in the subtlest lesions being missed from the image. Moreover, an expert radiologist must read the morphometric analysis yielded by the method. These limitations have caused researchers to turn their attention to new machine learning and deep learning methods. Around the early 2000s, machine learning methods grew in popularity given the increasing computing power of personal computers and their satisfactory performance in digital image analysis tasks. However, in recent times, they have been overshadowed by deep learning methods, which have been proven to achieve better performance.

This section introduces automatic FCD detection methods based on two types of deep learning algorithms: artificial neural networks (ANNs) and convolutional neural networks (CNNs). Although both methods can be categorized as deep learning, they are presented in two different subsections because of an important difference: to train ANNs, it is necessary to perform the image feature extraction process. In contrast, CNNs automatically extract their own high-level features.

##### Artificial Neural Networks (ANNs)

Using morphometry and a collection of neural networks to automatically extract brain sulci from MRIs, Besson et al. [54] quantified the special relationship between FCD lesions and sulci. To extract sulci, the authors initially segmented the image into GM, WM, and CSF maps and reconstructed the surfaces of the GM-WM and GM-CSF interfaces. Subsequently, they extracted the sulci by segmenting the GM-CSF interface after a “skeletonization” process. The sulci were labeled using a congregation of neural networks trained with manually delineated labels. For each control subject, the sulci that matched those related to each patient’s FCD were identified. Moreover, by checking their morphology and labeling, the authors ensured the best possible sulcal equivalence between subjects. The depth of a given sulcus was calculated by obtaining the sulcal bottom line, defined by the inner eastern edge. The depth of each point along this line was determined by calculating the shortest Chamfer distance from the outer cortical surface. Furthermore, the authors determined the mean and maximum sulcus depths for each subject from a database of 43 patients and 21 control subjects. Similarly, the mean depth of the lower portion of the sulcus in the vicinity of the lesion was also calculated. Lesions were classified as small or large according to a threshold defined by a measure of entropy. With a threshold of $T=3093\;{\mathrm{mm}}^{3}$, the method classified 21/43 cases as small FCD lesions. Among them, 17/21 cases (81%) had been excluded in the initial radiological evaluation. Likewise, 18/21 (86%) lesions were located at the bottom of the sulcus, 2 were related to the sulcal walls, and 1 was found at the crown of a gyrus.

Extending the scope of their method for separating small from large FCD lesions, Besson et al. [55] proposed a new approach to detect small FCD lesions on T1-weighted MRIs, which was based on surface features that model the morphometry and texture of FCD lesions. Automatic detection was performed by a neural network-based vertex classifier trained with manual labels. Vertex classification comprised a four-layer *feed-forward* ANN with six neurons in the first layer, four in the second, four in the third, and one in the output layer. From a database of 41 patients, 19 were found to have small FCD lesions according to the method described in [54]. These 19 patients were selected for the study along with 45 healthy subjects. The classifier detected lesions in 17/19 (89%) patients when the probabilistic threshold T=0.93 was selected, which provided the best balance between FPs and detection rate.

Subsequently, Besson et al. [56] complemented their methodology for detecting small FCD lesions by taking the classifier used in [55] as a starting point and adding a new classifier to reduce the number of FPs. Automatic detection was then performed using a two-stage classifier, where a second cluster-wise classifier (trained with manual labels from the first stage) was added to the neural network-based vertex-wise classifier. In the first stage, the same ANN algorithm was used, except for the modification of some hyperparameters. The clusters generated by this classifier were separated into FPs or FCD lesions by the second classifier based on global features. In the second stage, a fuzzy k-Nearest neighbor (fkNN) algorithm was used to determine the FCD lesion class membership. The database comprised the same 41 patients, 19 of whom had their lesions classified as small FCD according to the method described in [54]. These 19 patients were selected for the study, along with a cohort of 48 healthy subjects. The first classifier detected lesions in 18/19 (95%) patients, with a new probabilistic threshold T=0.87. The fkNN classifier, with k=20, retained 13/19 (68%) lesions and efficiently reduced the number of FPs.

To develop an automated FCD detection tool that addressed the challenges of pediatric patients, Adler et al. [57] sought to optimize its ability to find and quantify each area of the cortex in terms of how it differed from a healthy cortex. To this end, structural measurements were computed and post-processing methods were applied to quantify the number of radiological identifiers of FCD lesions. First, the authors dealt with the problem posed by well-established structural markers of FCD (e.g., cortical thickness and intensity), which may appear as normal regions and mask local abnormalities within an FCD lesion. Consequently, they normalized measurements across subjects, computed interhemispheric asymmetries between these measurements, and normalized the values for each vertex relative to control subjects. Next, structural markers common to FCD lesion regions (such as cortical thickness, GM-WM contrast, and FLAIR signal intensities) were quantified using *the doughnut method*. This method calculates the difference between an area of the cortex and its surrounding annulus at each vertex, highlighting the areas where these differences are most significant. In addition, the authors developed a local cortical deformation (LCD) measurement (based on the magnitude of the intrinsic curvature surrounding each vertex) as a more robust measurement of cortical shape. These structural markers and post-processing methods (i.e., LCD, interhemispheric asymmetry, and *doughnuts*) were combined with surface features common in FCD detection (i.e., cortical thickness, GM-WM boundary contrast, FLAIR signal intensity, curvature, and sulcus depth) to train an ANN to classify cortical regions into lesional and non-lesional vertices. The ANN consisted of a single hidden layer. The number of nodes was determined by principal component analysis (PCA) of the input features, using the number of components that accounted for more than 99% of the variance. The ANN was trained with a total of 28 features following the procedures described above. Moreover, the authors trained separate ANNs using individual features and data subsets to assess their discriminatory value. The study included 22 young patients with a radiological FCD diagnosis and 28 control subjects. Among the standard FCD features, FLAIR intensity was found to have the best discriminatory value (AUC = 0.83), followed by GM-WM contrast (AUC = 0.80), and cortical thickness (AUC = 0.63). Similarly, among the novel features proposed in that study, asymmetry in FLAIR intensity achieved the best discriminatory value (AUC = 0.87). Employing both novel and standard FCD features, the model was able to detect 16/22 (73%) FCD lesions. Using only standard FCD features, the model detected 13/22 (53%) FCD lesions, demonstrating that including the novel features helped to detect FCD more accurately.

Jin et al. [53] proposed an algorithm that automatically detects FCD using an ANN in order to assess the diagnostic value of surface-based morphometry (SBM) features in patients with histologically confirmed FCD type II. To train the ANN, the authors extracted six features from each vertex of the 3D cortical reconstruction: cortical thickness, GM-WM intensity contrast, curvature, sulcus depth, *doughnut* maps, and LCD. Additionally, the interhemispheric asymmetry of cortical thickness, GM-WM intensity contrast, and LCDs were also computed by subtracting the right hemisphere vertex values from the left hemisphere values to create the left hemisphere asymmetry map and vice versa. Moreover, separate ANNs were trained for each feature to assess their discriminatory value. A cross-validation strategy with k=5 was used to evaluate ANN performance, where vertices in the top 5% were identified and connected to neighboring clusters. Furthermore, a threshold was set by ROC curve analysis, so that vertices with values above the threshold were labeled as lesional and vertices below the threshold were labeled as non-lesional. Lastly, the detection was considered successful when the detected region matched the manual label. Their study included databases from three different radiology centers, for a total of 61 pediatric and adult patients—17 of whom were MRI-negative—and 120 control subjects. A test database with 15 patients and 35 healthy subjects was also included. Three factors were evaluated in terms of their effect on classifier performance: the number of training cases, the size of the control cohort, and separate training sets for each factor on each of the three databases. The algorithm performed best at a threshold of T=0.9, yielding a sensitivity of 73.7% and a specificity of 90%. In the case of the test data, the specificity was 91.4% in healthy subjects and 86.7% in patients. Regarding the analysis of the effect of the number of patients, it was found that the lower the number of patients included in the training, the lower the sensitivity and specificity of the model. A similar result was observed when the number of healthy subjects in the training was reduced. As for the effect of the type of scanner, it was found to have no significant impact on the performance of the algorithm.

Combining surface-based quantitative and multimodal features with machine learning, the authors of [58] built a model for automatic FCD detection to assess its clinical value. The study included structural and functional T1-weighted, T2-FLAIR, and PET sequences from 40 MRI-positive patients with histologically confirmed FCD type II and 33 control subjects with hippocampal sclerosis or epidermal cysts. Nevertheless, the authors were unable to assemble a database of healthy subjects that included PET images. After image preprocessing and normalization, they performed cortical reconstruction, which included extraction of non-brain tissue, segmentation of the GM and WM structure, and tessellation of the GM-WM boundary. Subsequently, several types of features were extracted: morphological (GM-WM contrast intensity, LCD, cortical thickness, mean curvature, sulcus depth, intensity *doughnut*, and thickness *doughnut*), intensity (FLAIR intensity at a different level from cortical depth and FLAIR *doughnut*), and metabolic (hypointense PET and PET asymmetry). Automatic detection was performed using an ANN classifier trained with the above features on vertices labeled as lesional and selected vertices labeled as non-lesional. PCA was applied to reduce input dimensionality and speed up training. The database was divided into 70% for training, 15% for validation, and 15% for testing. The training was carried out using cross-validations (k=5) with 100 iterations, finding lesions whose location matched the manual label in 31/40 patients, for a detection rate of 77.5% and a sensitivity of 70.5%.

Wagstyl et al. [59] evaluated the feasibility of incorporating MRI-trained deep learning algorithms into the planning of stereoelectroencephalography (sEEG) cathode implantation in pediatric patients with FCD. Their study analyzed the degree of colocalization between automated lesion detection and the seizure onset zone detected by sEEG. It included T1-weighted and FLAIR sequence images from a cohort of 34 MRI-positive and 34 sEEG patients. Both databases consisted of subjects between 3 and 18 years of age. In addition, a group of 20 control subjects with no neurological history was also included. To train the deep learning algorithm, the authors extracted the following morphological and intensity features (computed per vertex across the cortical surface): cortical thickness; GM-WM contrast intensity; curvature; sulcus depth; intrinsic curvature; and FLAIR signal intensity sampled at 25%, 50%, and 75% of the cortical thickness and at the GM-WM boundary. Lastly, interhemispheric asymmetry maps were created. The ANN architecture featured a hidden layer, whose number of nodes, computed by PCA on the input features, was the minimum number of components that accounted for at least 99% of the variance. The algorithm found FCD lesions in 25/34 MRI-positive patients and detected no lesions in any of the 20 control subjects, which means a sensitivity of 74% and a specificity of 100%. Among the 34 patients who underwent sEEG, an FCD lesion was identified in 21 of them. Compared to the patients who underwent sEEG, the automatically predicted lesion matched the sEEG lesion in 13/21 patients, for a 62% colocalization. This suggests that the automatic algorithm may help to avoid surgical electrode implantation.

To improve existing methods for automatic FCD detection, Ganji et al. [11] designed and implemented a computerized diagnostic system to identify FCD lesions, assuming that these lesions can be accurately detected and localized using morphometric and machine learning methods. To evaluate specificity, the authors employed structural images (T1-weighted and T2-FLAIR sequences) of 30 patients with confirmed FCD type II and 28 control subjects. The images underwent cortical reconstruction, followed by the extraction of various morphological and intensity-based features. To this end, the cerebral cortex was divided into 34 regions per hemisphere. In addition, features such as average cortical thickness, Gaussian curvature, mean curvature, intrinsic curvature index, and folding index, as well as descriptive statistics, including mean, minimum, maximum, range, standard deviation, and signal-to-noise ratio for intensity contrast, were calculated for each region. The classification process served two purposes: First, to determine whether a region was lesional or non-lesional, and second, to detect the lesion area. The ANN classifier was trained for 30 epochs, with 70% of the data being used for training, and the remainder being divided equally for validation and testing. Lesion location involved determining the (i) brain hemisphere (right or left) and then the (ii) brain lobe (frontal, temporal, parietal, or occipital) where the lesion was located. After training the algorithm for 30 epochs, its average sensitivity, specificity, and accuracy were 96.7%, 100%, and 98.6%, respectively. When tested on 10 MRI-negative patients, its accuracy was 91.3%. Finally, its accuracy in identifying the brain hemisphere and lobe where the lesion was located was 84.2% and 77.3%, respectively.

Seeking to obtain a robust, automated classification of FCD, David et al. [60] developed a classifier by integrating information from three types of morphometric maps together with the tissue segmentation results from an ANN. They tested the generalization and performance of the classifier on a large, independent dataset of patients with histologically confirmed FCD. First, the authors performed a morphometric analysis of the images to extract extension, junction, and thickness maps. Then, using a database of T1-weighted sequence MRIs from 113 pediatric and adult patients and 362 healthy adult subjects, the following feature maps were extracted to train the ANN: a normalized T1-weighted image, morphometric maps (extension, junction, and thickness), smoothed morphometric maps, tissue maps (GM, WM, and CSF), smoothed tissue maps, a brain map, and a GM-WM boundary map. The ANN was designed with a two-layer architecture, featuring five sigmoid neurons in the hidden layer and two softmax neurons in the output layer to distinguish between two classes: dysplastic and non-dysplastic voxels. For the classification of a specific voxel, the values of the 15 feature maps were read at the corresponding voxel and used as input for the ANN. Training performance was evaluated by cross-validation with 10 iterations. The validation database included 60 pediatric and adult FCD patients and 76 healthy adult subjects. In the performance evaluation using the training data, the ANN detected on average 98.8/113 FCDs, resulting in a mean sensitivity of 87.4% and a mean specificity of 85.4%, which corresponds to an accuracy of 85.9%. The performance of the model was further assessed using an independent dataset consisting of 58 patients with FCD and 70 healthy subjects. In this evaluation, the method detected 47/58 FCDs, achieving a sensitivity of 81%. Regarding healthy subjects, it detected lesions in only 11/70 subjects, indicating a specificity of 84.3% for the independent dataset.

The objective of the study conducted by de Freitas et al. [13] was to develop a classification method based on morphological feature extraction, in which a machine learning algorithm could recognize patterns and assist healthcare personnel in detecting FCD lesions. In particular, the authors introduced a classification method based on a multilayer perceptron (MLP). In addition, they trained multiple traditional classifiers and compared their performance to select the most effective one. Their study also made novel contributions, including the development of an extended feature vector, the creation and evaluation of an ensemble of machine learning methods for FCD detection, and the assessment of patient-level classification. After a preprocessing and brain extraction phase, 20 morphological and texture features were extracted to feed the classification algorithms. The descriptor vector included eight Haralick texture features (energy, entropy, correlation, contrast, cluster hue, homogeneity, cluster salience, and Haralick correlation) along with intensity level and cortical thickness, which is a morphological feature. These features were then compared to a template or map representing a healthy cerebral cortex, and the *z-score* of each feature was calculated, resulting in a feature vector of size 20. Additionally, ten new features were generated by summing the ten features with the highest impact and the ten with the lowest impact. The effects of each were measured using the *SelectKBest* algorithm of *scikit-learn* to achieve a feature vector of size 30. The classifiers employed by the authors, which were implemented using empirically obtained hyperparameters, included (a) XGBoost, (b) KNN, (c) decision trees, (d) random forests, (e) MLP, (f) Naïve Bayes, (g) AdaBoost, and (h) SVM. The proposed MLP classifier consisted of a hidden layer with 100 neurons using the ReLU activation function and the Adam optimizer and was trained for 5000 epochs with a learning rate of 0.001. Seven-fold cross-validation was used to ensure a minimum of two patients for testing. The study employed MRIs from 15 patients. At the voxel level, the MLP classifier achieved the best performance with an average accuracy of 96.81%, a precision of 97.12%, and a sensitivity of 96.47%. At the patient level, the SVM classifier demonstrated the best accuracy with 90.64%, followed by the AdaBoost classifier with 90.22%, and the MLP classifier with 89.3%, which obtained good results at both the voxel and patient levels.

##### Convolutional Neural Networks (CNNs)

Taking advantage of the ability of CNNs to extract features automatically, Gill et al. [61] proposed an algorithm that requires minimal data preprocessing. They used T1- and T2-sequence MRIs for training the algorithm and validated it, for the first time according to the authors, on a database from seven different radiology centers around the world. To eliminate the class imbalance problem inherent to FCD lesions, where the number of healthy tissue examples is significantly higher than that of malformations, the authors suggested training the algorithm with patches extracted from the MRIs. They created a training dataset that included an equal number of examples from both classes and selected only those where the hyperintensity in the FLAIR sequence exceeded a predefined threshold. The classifier designed by the authors consisted of two identical, cascaded CNNs with independently optimized weights. The first CNN was trained to maximize the identification of lesional voxels, whereas the second CNN was trained to reduce the number of FPs while maintaining optimal sensitivity. Each CNN comprised three stacks of convolutional and *max pooling* layers, each with 48, 96, and 2 filters, respectively. Rectified linear unit (ReLU) activation was used in the first two layers, while softmax activation was employed in the output layer to obtain a binomial distribution of the estimated classes. To train the algorithm, the database, which consisted of a cohort of 40 subjects with histologically confirmed FCD, was divided into 75% for training and 25% for validation. The weights were randomly initialized, and the binary cross-entropy (BCE) was used as the cost function. During training, the model underwent five-fold cross-validation for five epochs. It was evaluated on an independent cohort comprising 67 adult and pediatric subjects with histologically confirmed FCD from six radiology centers, 38 healthy control subjects, and 63 control subjects with temporal lobe epilepsy (TLE). Cross-validation during the training of the CNNs resulted in a sensitivity of 87±4%, with an average detection of 35/40 lesions and a specificity of 95% in healthy subjects and 90% in subjects with TLE. When evaluating the model on test subjects from the seven different centers, its sensitivity was 91%, detecting 61/67 lesions, with 3±2 FPs observed in 47/67 cases.

In [62], the authors introduced the first CNN algorithm for the detection and segmentation of FCD. The proposed algorithm was trained exclusively with MRIs in the FLAIR sequence because the hyperintense regions caused by the presence of FCD are more visible in this sequence. The methodology developed by the authors comprised the following steps: preprocessing (noise reduction and skull extraction), FCD segmentation (U-Net architecture), and post-processing (filtering). The U-Net architecture used in the study took input images with a size of 256×256. In the contracting path, each block consisted of two convolutional layers with 3×3 filters, followed by a ReLU activation function, *batch* normalization, and a *max pooling* operation. To compensate for the loss of spatial information due to *max pooling*, the number of filters in the convolutional layer of the subsequent blocks was doubled. In the expanding path, the spatial dimensions of the feature maps were increased through transposed convolutions. Then, the high-resolution feature maps from the contracting path were concatenated with the resulting unsampled image. After each concatenation, a *dropout* layer was employed to reduce the effect of overfitting. Finally, a 1×1 convolutional layer with sigmoid activation was used, producing an image with the exact dimensions as the original, which was subsequently thresholded to obtain a binary segmentation mask. The proposed algorithm was trained for 100 epochs, and the cost function was a combination of the BCE and the Dice coefficient. The Adam optimizer was employed to update the weights with a learning rate of 0.001, which was reduced by a factor of 0.4 when the validation cost did not improve its performance. Moreover, real-time data augmentation techniques (rotation, zoom, and horizontal rotation) were used during model training. Finally, to reduce the number of FPs, an average filter with a *stride* of 1 and a size of 5×5 was applied. For the experiments, the authors employed a database comprising 43 subjects with refractory epilepsy due to FCD. It included 17 MRIs obtained with a 1.5T scanner, each with a size of 256×224 and a number of slices ranging between 144 and 192, as well as 26 MRIs acquired with a 3T scanner, each with a size of 512×512 and 320 slices. The algorithm was trained with 2D slices from each MRI volume. The dataset was divided into 80% for training and 20% for testing, resulting in 5680 images for training, 1419 for validation, and 1780 for testing. The proposed model achieved a detection rate of 82.5% (33/40), with an accuracy of 80.69% and a Dice coefficient of 52.47%.

In their study, Wang et al. [63] presented a technique for the automatic detection of FCD using MRI and deep learning with a patch-trained CNN architecture. The authors employed T1-weighted MRIs from a cohort of ten patients with confirmed FCD, ten healthy subjects, and ten subjects with TLE as the control group. These images underwent various preprocessing steps, including brain extraction, intensity correction, and normalization. Then, rectangular patches of the brain cortex on the axial axis of each MRI were extracted by first extracting mask patches from the GM probability map and then applying the mask to the preprocessed MRI of the brain. A total of 129,780 patches (4326 patches/image) were extracted, 2302 labeled as FCD and 127,478 as non-FCD. To address the class imbalance problem and thus increase the number of FCD labeled patches, data augmentation was performed using the upsampling method, resulting in 36,786 FCD patches. The same number of non-FCD labeled patches was randomly selected to balance the training data. The dataset was divided into 70% for training, 15% for validation, and 15% for testing. The proposed CNN architecture consisted of five convolutional layers, one *max pooling* layer, and two fully connected (FC) layers. Each convolutional layer comprised three operations: convolution, *batch* normalization, and ReLU activation. Following the convolutional layer, a *pooling* layer was applied, where the number of parameters to be trained was reduced by downsampling using a *max pooling* layer with a 2×2 filter. Then, the dimensions of the feature maps were reduced by means of a *flatten* operation, resulting in 8448×1 feature vectors. Finally, the feature vectors passed through the 256 nodes of the first FC layer, and the output was fed into the second FC layer, which consisted of two nodes. The final activation employed by the authors was a softmax layer, which provided the probability of each class. The cost function used for training was the cross entropy function for multiclass classification, which was optimized using stochastic gradient descent (SGD) with momentum. With the proposed method, the authors obtained an accuracy of 0.941 and an area under the ROC curve of 0.985 during training, indicating good algorithm performance. Cross-validation was employed for validation, and metrics such as sensitivity, specificity, accuracy, and the Dice coefficient were calculated. The proposed method detected 9/10 FCD lesions in the database of subjects with lesions, yielding a sensitivity of 90%. When evaluated on the control subjects, its specificity was 85%, its accuracy was 88%, and the mean Dice coefficient was 0.78.

Using a CNN, the authors of [12] aimed to classify and localize FCD lesions in the MRIs of patients with epilepsy via FCD simulation. For training, the authors acquired 636 MRIs from 89 healthy pediatric and adult subjects and locally added a bright region to these images to simulate the FCD lesion area. Three training datasets were created: the first with a 1.5 times intensity magnification in the simulated lesion area, the second with a 1.2 times magnification, and the third with a 1.1 times magnification. Each dataset contained 1272 images, with 636 labeled as positive samples and the other 636 as negative samples. For testing, the authors used a database comprising 53 MRI volumes, of which 18 were from 10 pediatric and adult patients with FCD and 35 were from 6 healthy adult subjects. The CNN architecture employed in the study consisted of four convolutional layers with sigmoid activation, each followed by an *average pooling* operation. The last layer was fully connected to the activation functions of the neurons in the preceding layer, whose activation was also the sigmoid function. The mean squared error was used as the cost function, and it was optimized by the gradient descent method with a learning rate of 1 for 2000 epochs. After training the proposed model using the three generated datasets, the authors evaluated its performance. For the first training dataset (with a 1.5 times magnification), the classification accuracy was 100% and the localization accuracy was 65% during training and 67.92% and 16.67%, respectively, during testing. For the second training dataset (with a 1.2 times magnification), the classification accuracy was 100% and the localization accuracy was 89% during training and 79.25% and 71.43%, respectively, during testing. The best results were obtained with the third dataset (with a 1.1 times magnification), with a classification accuracy of 100% and a localization accuracy of 92% during training and a classification accuracy of 92.45% and a localization accuracy of 92.86% during testing.

In a subsequent study, Feng et al. [64] developed an activation maximization and convolutional localization (AMCL) method based on a CNN to detect and segment FCD lesions in FLAIR images from MRI-negative patients with histologically confirmed FCD. The proposed method consisted of five steps: (a) image preprocessing, (b) CNN design and training, (c) generation of pattern image blocks (PIBs) using the trained CNN, (d) lesion localization and segmentation using the average PIB with convolution, and (e) quantitative evaluation of the method’s performance. In the image preprocessing phase, several steps were applied, including brain extraction, resizing of the images to 256×256 pixels, and normalization. Then, positive blocks with a size of 28×28 pixels were extracted from the lesion area, while negative blocks of the same size were extracted from the normal area for training. A total of 126,684 negative and 148,428 positive blocks were obtained for CNN training. The CNN training phase involved training a CNN named Net-Pos, which consisted of two convolutional layers with ReLU activation, two *max pooling* layers, and two fully connected layers. The log-likelihood function was chosen as the cost function, which was optimized using SGD with momentum and a learning rate of 0.01. The data were divided into 80% for training and 20% for testing. To obtain the PIB for the third phase of the method, the authors applied the concept of activation maximization by gradient descent. This involved feeding the trained model with noise or a random image, calculating the gradients of a specific neuron in a specific layer with respect to the noisy image to maximize the neuron, and iteratively updating each pixel in the noisy image to obtain the pattern image or PIB. The average PIB was calculated and used for the subsequent convolutional pattern-matching operation. In the last phase, the performance of the proposed method was evaluated using completeness, specificity, and accuracy techniques. The method was tested using 18 FLAIR-negative lesion images from 12 patients, which resulted in a subject-level sensitivity of 83.33% (15/18 lesions detected). Its segmentation performance, which was measured with the Dice coefficient, was 52.68%.

Using improvements to existing deep learning methods, particularly the technique proposed in [63], the authors of [65] attempted to solve the problem of automatic detection of FCD in MRIs. The training dataset included labeled MRIs from 15 subjects with FCD, 15 unlabeled MRIs, and MRIs from 17 healthy control subjects. For this study, an expert radiologist provided the labels using 2D bounding boxes per view (axial, coronal, and sagittal). The authors employed the technique proposed in [63] as the base model, which consists of four steps: (a) preprocessing, (b) brain patch extraction, (c) classification by deep learning, and (d) post-processing. Preprocessing involved various tasks such as brain extraction, normalization, and generation of *ground truth* regions forming (3D) parallelepipeds from the 2D bounding boxes. According to the authors, patch extraction is a fundamental part of the algorithm, which is why they evaluated the models based on the probabilities assigned to the patches. For such evaluation, they introduced the *top-k* scoring method. This metric considers the *k* best patches by probability and estimates whether at least one of them intersects with the *ground truth* region, indicating that the detection was successful. Although the base model achieved a *top-k* score of 0.2, the authors made several improvements to enhance its performance. For instance, (1) 3D ellipses replaced the parallelepipeds with masks. (2) A ‘soft’ labeling was implemented, assigning values close to 1 and close to 0 based on the degree of overlap between the ellipse and the *ground truth* region, rather than just using zeros and ones. Also, (3) the size of the patches was changed from 16×32 to a new size of 24×40 because their original size was not sufficient to cover all brain regions. Additionally, (4) coronal and sagittal axis slices were included, using those slices that intersected in the center with their respective axial axis slice and their mirrored neighbors. (5) Since in 8/15 patients, the FCD lesion was located in the temporal gyrus and in 7/15 patients in the non-temporal region, and given the differences between these two regions, different models were trained for each of the two groups of subjects. Finally, (6) to take advantage of the 15 unlabeled MRIs, the authors pre-trained an *autoencoder* with a *3 heads–3 tails* architecture. Each slice of each axis underwent its own convolution, concatenating the results and then performing a linear operation. According to the findings, the model with the best *top-k* score (i.e., the best performance) detected 11/15 FCD cases, resulting in a sensitivity of 73.3%.

In an attempt to overcome the limitations of the U-Net network implemented in [62], Thomas et al. [66] presented a new version of it. This version incorporates the optimal location of attention gates, the correct choice of feature maps for the gates’ activation signal, and the optimal location of residual paths (ResPath) within the proposed architecture for FCD segmentation. The authors believed that the drawbacks of the original U-Net architecture [62] could be mitigated by adding the following design features: (a) attention to salient features (to prevent the loss of the finest features between the intermediate and final layers, thus allowing the model to focus on specific regions of the image), and (b) robustness to scale and reduction of the semantic gap (because images used for tasks involving segmentation of, for instance, tumors, lesions, and organs contain varying scales of ROIs. Furthermore, the semantic gap increases the learning discrepancy and is detrimental to model performance). To solve the scale invariance problem, the authors employed multi-residual blocks. First, these blocks reduced the high memory requirements of the 5×5 and 7×7 kernels by using 3×3 kernel chains in series. Second, they increased the number of filters of the three successive layers in steps, thus preventing the memory overhead of the previous layers from propagating to the next layers. To lessen the effects of the semantic gap, the authors adopted the ResPath procedure, since they believed that a very deep model was not desirable given the small size of the FCD lesions. They proposed an architecture with a depth of 3, where the encoder block at depth 0 extracted low-level features, while the mirrored decoder layer at depth 0 received high-level feature maps. Since this discrepancy was more pronounced at depth 0, the authors confined the use of ResPath and introduced nonlinearity in the feature maps to reduce the semantic gap. For its part, the attention block consisted of an attention activation signal taken from a deeper multi-residual block, which contained the finest features and informed the attention block which features were the most relevant to focus on. This resulted in a feature map that emphasized the salient features of the input image. The algorithm was trained using FLAIR sequence MRIs from 26 patients, with 80% of the dataset used for training (including 15% for validation) and 20% for testing. Five-fold cross-validation was employed for model validation, and real-time data augmentation was applied to reduce overfitting. The training cost function was represented as the sum of the Dice coefficient and the BCE cost, and the Adam optimizer was used with a learning rate of 0.001. According to the results, the proposed method achieved a detection rate of 92% at the patient level. Additionally, with a Dice coefficient of 76.62%, the proposed model showed an improvement in the Dice coefficient (at the region level) of 5.09%, 6.71%, and 6.40% with respect to the base model [62] and the models proposed in [74,75]. However, the authors acknowledged that the model falls short of accurately identifying the true extent of an FCD lesion.

In their study, Gill et al. [67] employed a CNN to test the hypothesis that a deep learning algorithm, validated on a multi-center database, can detect FCD lesions in MRI-negative patients. This study included retrospective cohorts from nine tertiary epilepsy centers worldwide, comprising patients with histologically confirmed FCD lesions and their corresponding T1-weighted and FLAIR sequence MRIs. The primary center consisted of 62 patients with FCD, along with a control group of 42 healthy subjects and 89 subjects with TLE (49 of whom exhibited hippocampal sclerosis visible on MRI, and 40 with no visible signs). The cohort from the remaining eight centers comprised 86 patients with FCD. The FLAIR sequence MRIs of each individual were intensity-normalized by calculating the *z-score*, and intensities below the 10th percentile were discarded. This resulted in a mask containing the voxels that belonged to the GM-WM interface. From this mask, 3D patches were extracted from the co-registered T1-weighted and FLAIR sequence MRIs, which were then used as input to the CNN. The authors designed a cascade classification system with two CNNs, where the output of the first CNN (CNN-1) served as input to the second CNN (CNN-2). CNN-1 aimed to maximize the detection of lesional voxels, while the goal of CNN-2 was to reduce the number of FPs while maintaining sensitivity. The algorithms were trained with the 3D patches from the T1-weighted and FLAIR sequence MRIs obtained from the nine radiological centers using leave-one-center-out cross-validation. Additionally, a test dataset, utterly independent of the training dataset, was created with 23 FCD patients (70% of whom had negative MRI results) selected from centers 1 and 2. The performance of the model was evaluated at both the patient and voxel levels. At the patient level, the training sensitivity was 93% (137/148 FCD lesions detected), with an average of 6 FPs per patient. In the independent cohort, the sensitivity was 83% (19/23 FCD lesions detected), with an average of 5 FPs per patient. The specificity was 90% in healthy subjects and 89% in subjects with TLE. At the voxel level, the area under the ROC curve was 0.83, indicating that the model achieved high sensitivity and specificity.

In [68], a new 3D CNN with *autoencoder* regularization for the automatic detection and segmentation of FCD was developed and prospectively validated on routine MRIs. The training dataset consisted of 201 T1-weighted and FLAIR sequence MRIs and 172 T2-weighted MRIs. In addition, 100 MRIs from healthy subjects and 50 MRIs from patients with different neurological pathologies were included. The test database consisted of 100 consecutive MRIs that were routinely obtained from patients with FCD. Both databases comprised MRIs from both adult and pediatric subjects. Image preprocessing involved three stages: (i) segmentation of brain tissues (GM, WM, and CSF), (ii) morphometric analysis (creation of joint and spread images), and (iii) final input preparation for the CNN (cropping, padding, and bias correction). The proposed CNN combines several techniques. It begins with a variation of the encoder–decoder architecture and then applies a variational self-encoder as a separate branch of the decoder and as a regularization method. The authors also explored other techniques such as transfer learning and dense connections but did not observe improvements in the results. The key element of the proposed architecture is the residual block, which consists of two convolutions, a ReLU activation function, and group normalization with the same number of groups as channels. Also, there is a skip connection at the end of the block. The encoder comprises three levels. Each level has two residual blocks (except for the first level, which only includes one residual block), with the first one increasing the number of channels by a factor of two. Convolutions with *stride* and positional normalization layers that capture the structural information of the image are used as subsampling layers. The decoder uses only one residual block at each level, and after upsampling with trilinear interpolation, its output is partially added to the *skip connection* of the corresponding level in the encoder. Then, the *moment shortcut* is employed to reverse the positional normalization, and a sigmoid activation function is applied at the end to obtain a probability map ranging between 0 and 1. The *variational* decoder serves as a regularization mechanism for the model and does not influence segmentation. It uses three convolutions and two linear upsampling layers. For training and validation, the 201 T1-weighted and FLAIR sequence MRIs were divided into subsets of 80% for training and 20% for validation. Five-fold cross-validation was employed, and the Adam optimizer was used with an initial learning rate of 0.0001 for 80 epochs. Data augmentation techniques (such as rotation and zoom) were also implemented. The cost function used to train the algorithm was a combination of the Dice cost function, the mean squared error, and the Kullback–Leibler divergence. To measure algorithm performance, the Dice coefficient, accuracy, sensitivity, and *F-score* were calculated. The best training results were obtained when the algorithm included 100 MRIs of patients with pathologies other than FCD. It yielded a sensitivity of 70.1%, an accuracy of 54.3% in detecting FCD, and a Dice coefficient of 34.1% in segmentation. In the prospective evaluation, a sensitivity of 77.8% and a specificity of 5.5% were obtained.

In [69], the authors compared the performance of two types of models in classifying and segmenting FCD lesions in pediatric patients with refractory epilepsy. The first model was a special type of CNN known as a fully convolutional network (FCN), and the second model was a multi-sequence FCN. The FCN-based model was trained separately using slices from T1-weighted, T2-weighted, and FLAIR sequence MRIs, while the multi-sequence model was trained by simultaneously incorporating the combined information from slices of T1-weighted, T2-weighted, and FLAIR sequence MRIs. The database comprised T1-weighted, T2-weighted, and FLAIR sequence MRIs from 80 pediatric patients with FCD (56 MRI-positive and 24 MRI-negative) and 15 healthy control subjects. Image preprocessing included brain extraction, manual mask delineation, and image resizing by adding or removing background voxels. The architecture of the FCN comprised four two-layer blocks of convolutions and *pooling*. The fully connected layers were then replaced with three upsampling blocks, each with one convolutional layer and one upsampling layer. Each layer in the architecture was followed by a ReLU activation function. Moreover, skip layers were used to enhance the output of the upsampling layers by adding the respective downsampled feature map from the encoder stage. Additionally, *batch* normalization and *dropout* were implemented to avoid overfitting. The architecture of the multi-sequence FCN, for its part, consisted of two networks working together to extract features from all sequences, combine them, and predict a segmentation map. The first network was trained with different sequences and labels at the subject level, and the output of the last convolutional layer was used as input to the second network. The second network (an FCN) took the three feature maps from the T1, T2, and FLAIR sequences, concatenated them, and upsampled the input to arrive at the original slice size and generate the segmentation map. Like the architecture of the first FCN, this one included four blocks of three convolutional layers. In each block, the first two convolutions had filters with a size of 3×3, a *stride* of 1, and a *padding* of 0, and the last convolutional layer had a filter with a size of 2×2 and a *stride* of 2. The blocks were followed by a 1×1 convolutional layer and then three fully connected layers with *dropout* and *batch* normalization to classify the extracted features into the ‘lesional’ and ‘healthy’ classes. All layers used the ReLU activation function except the last one, which employed the sigmoid activation function. The cost function used by the authors was the BCE, which was optimized using SGD. The upsampling FCN consisted of two convolutional layers with 128 filters with a size of 3×3, a *stride* of 1, and a *padding* of 0. This was followed by three upsampling blocks: the first two blocks included a 2× upsampling layer and a 3×3 convolutional layer followed by *batch* normalization, and the last block consisted of a 4× upsampling layer followed by a 1×1 convolution. A ReLU activation function was employed for all layers except for the last one, which used the sigmoid function. The *leave-one-out* technique was employed to train and evaluate the model. After obtaining the model predictions, the authors divided the number of pixels predicted as lesional over those predicted as healthy to determine whether the subject belonged to the lesional or healthy class. This ratio was thresholded using a threshold value obtained from the ROC curve to assign a label to each pixel. According to the findings, the FCN algorithm demonstrated its best performance in the T1-weighted sequence, with a detection rate of 91% (52/56 MRI-positive cases and 21/24 MRI-negative cases) and a Dice coefficient of 56%, followed by the T2 sequence. However, the multi-sequence model performed best, as it detected 96% of cases (55/56 MRI-positive cases and 22/24 MRI-negative cases) and obtained a Dice coefficient of 57%. For testing, the authors selected a cohort of six patients, in which the multi-sequence algorithm detected 4/6 cases, resulting in a sensitivity of 67% and a Dice coefficient of 58%. Table 6 shows the results obtained in the different studies that have used CNN architectures.

This section included the state-of-the-art deep learning-based methods: ANNs and CNNs. They have been widely used in recent years due to advances in computing power. Their demonstrated performance has surpassed that of more traditional machine learning techniques without needing hand-crafted feature extraction from the image. According to our review, ANN-based studies in this section have achieved good results in terms of FCD detection and segmentation. However, recently developed CNN-based algorithms, such as U-Net and Attention U-Net, have reached state-of-the-art performance in FCD detection and segmentation, despite the fact that these are data-intensive algorithms and the scarcity of MRI data from FCD patients.

## 3. Discussion and Conclusions

MRI images can be used effectively to study focal cortical dysplasia (FCD). However, in many cases, there is no evident abnormality in MRI images, or the visual interpretation is tedious and subjective, prone to inter-observer variations. An automated system to detect FCD can help clinicians to make decisions regarding surgical resection, which can improve patients’ quality of life by preventing the occurrence of epilepsy.

We presented an overview of methods for FCD detection—visual, semi-automatic, and fully automatic (based on machine learning and deep learning)—highlighting the significant progress that has been made in this field. This classification of techniques follows the way they were chronologically presented in the literature. First, mathematical models (which are morphological methods) were introduced to enhance feature maps and thus improve visual FCD detection. Subsequently, machine learning methods (especially SVMs) gained momentum. Currently, FCD can be automatically detected by deep learning methods that use neural networks based on U-Net and *autoencoders*.

However, large datasets are crucial for data-driven methods (such as the more recent deep learning approaches). Several papers have reported that an insufficient number of data has been a limitation to the generalization of their methods. From a summary of the information about the datasets used by the studies included in this review (Table 1), we found that there is no benchmark dataset that allows a direct comparison between the different proposed approaches to FCD detection. Therefore, there is a need for publicly labeled datasets that comply with anonymization conditions. Nevertheless, data augmentation can be combined with generative models to produce large labeled datasets—so far, this has been conducted only rarely and requires expert supervision. Another possible strategy to overcome this difficulty is to adopt a distributed approach, known as federated learning, which allows multi-institutional collaboration to train automatic models without sharing the actual data.

Regarding the performance metrics, there is no consensus regarding a single set of measures to validate the results. Since most studies approach FCD detection as a classification task, some traditional supervised metrics, such as accuracy, sensitivity, specificity, and F1 score, are used as quantitative assessments. In addition, segmentation metrics, such as the Dice coefficient or ROC curves, are used to quantify the degree of overlap between the obtained result and a true mask. Again, the lack of a benchmark dataset does not allow a direct comparison in terms of classification and segmentation between studies. In this sense, it is also necessary to have a baseline result of human performance with different experience levels, making it possible to establish reference values for each metric.

The integration of automated tools into FCD detection and localization holds great promise for improving diagnostic accuracy and efficiency. In the near future, these tools will be incorporated as graphical functions in medical image-viewing software used by neurologists and radiologists, thus facilitating the identification and delineation of FCD regions during routine clinical practice. This integration will enhance the diagnostic workflow and aid in the precise planning of surgical interventions, ultimately leading to improved patient outcomes.

Furthermore, ongoing advances in MRI technology offer exciting prospects for FCD diagnosis. New MRI sequences and the availability of higher-field-strength scanners (such as 5 T, 7 T, and above) can potentially enhance spatial resolution and anatomical detail. These advances will contribute to a more comprehensive visualization of FCD lesions, enabling larger multi-center and multi-scanner datasets. The application of such advanced imaging techniques will undoubtedly enhance the accuracy and reliability of FCD diagnoses.

To summarize, access to more data, tools integrated into the clinical practice, and new sequences with the potential for acquisition with high magnetic field scanners will allow the use of advanced artificial intelligence techniques. Alternatively, learning models can be trained to solve tasks of a similar nature, for which massive and public datasets are available, e.g., brain tumor detection. Subsequently, the knowledge gained by these models can be leveraged to fine-tune them to solve the FCD detection task.

## Figures and Tables

**Figure 1 sensors-23-07072-f001:**
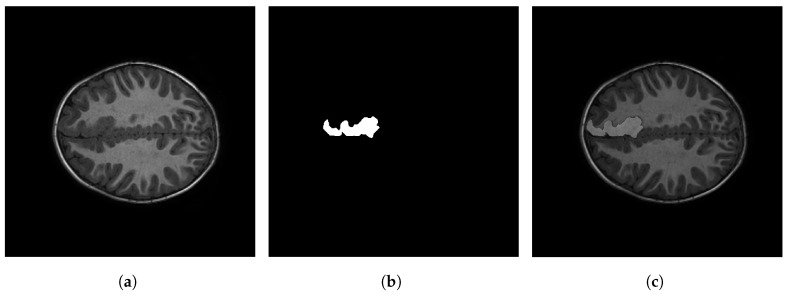
Example of one MRI image on axial view with a focal cortical dysplasia segmented by an expert. (**a**) MRI slice in axial view. (**b**) Mask or manual segmentation. (**c**) Manual segmentation on MRI.

**Figure 2 sensors-23-07072-f002:**
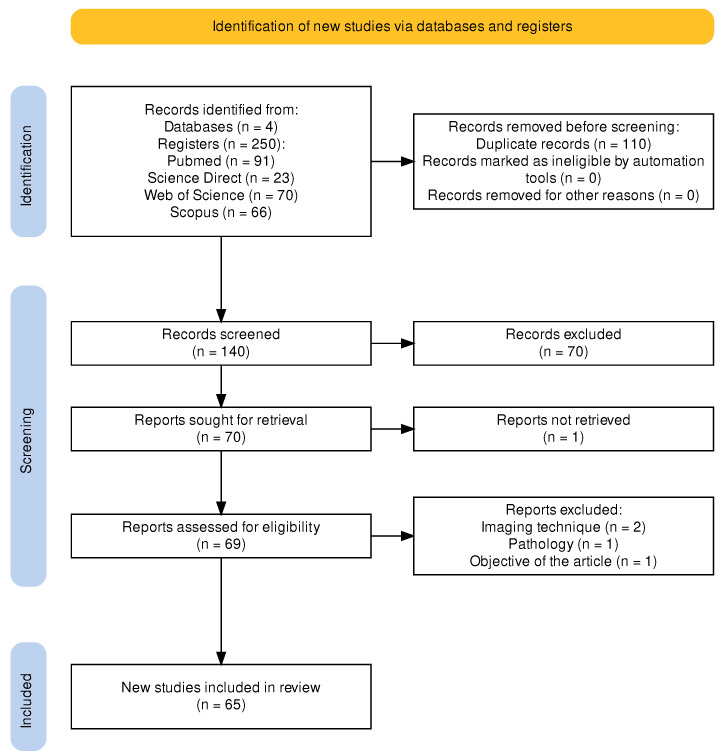
Flowchart of the PRISMA protocol applied in this systematic review. It includes three main steps: (1) identifying studies in selected databases, (2) screening identified studies, and (3) reporting on included studies.

**Figure 3 sensors-23-07072-f003:**
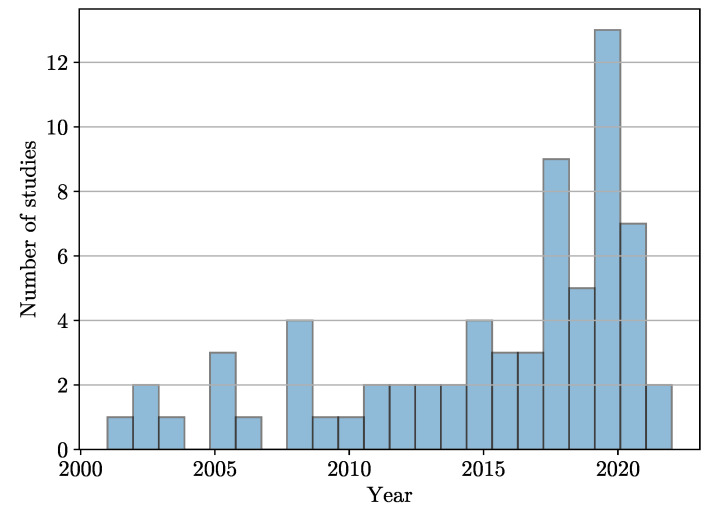
Distribution of FCD studies found with the PRISMA methodology.

**Figure 4 sensors-23-07072-f004:**
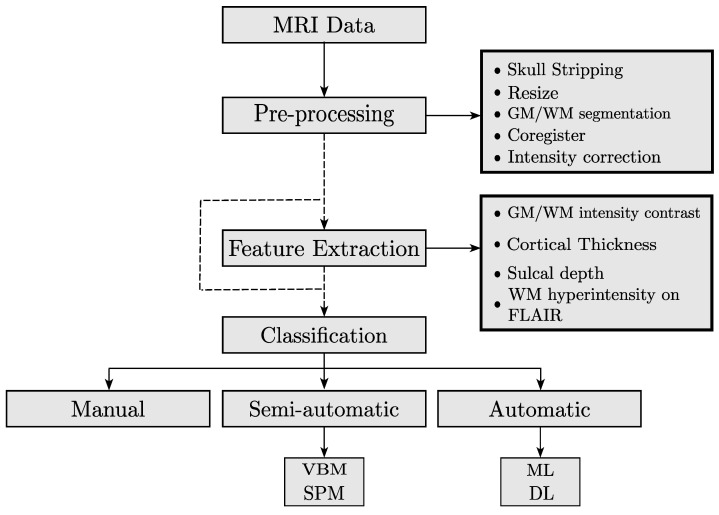
General framework for FCD classification. The preprocessing step aims to enhance the image for manual or automatic identification of FCD. Furthermore, some studies lack the feature extraction step because the images can be used as features in deep learning approaches.

**Table 1 sensors-23-07072-t001:** Dataset descriptions from the studies included in the review.

Study	Patients	Age Range	Sequence	Magnetic Field	Availability of the Dataset
[8]	12	15 ± 8	T1, T2, FLAIR	1.5 T	—
[9]	31	range from 14 to 51	T1	3 T	—
[11]	30	range from 1 to 46	T1, FLAIR	1.5 T	available upon request
[12]	10	range from 5 to 28	—	—	—
[13]	15	—	T1, T2, FLAIR	—	—
[15]	4	32 ± 13	dMRI	7 T	available upon request
[16]	14	—	T1	1.5 T	—
[17]	29	median 20 (IQR 13–29)	T1	1.5 T–3 T	—
[18]	20	range from 3 to 43	T1	1.5 T	—
[19]	33	10 ± 4	T1, FLAIR	3 T	—
[20,21]	24	24 ± 8	T1	1.5 T	—
[22]	8	—	T1	3 T	—
[23]	15	range from 15 to 53	T1, T2, FLAIR	3 T	—
[24]	9	range from 18 to 36	T1	3 T	available upon request
[25]	7	range from 14 to 51	T1	1.5 T	—
[26]	8	—	T1	3 T	—
[27]	20	range from 17 to 59	T1, T2	3 T	—
[28]	10	—	T1	3 T	—
[29]	16	26.4 ± 6.2	T1	3 T	—
[30]	10	36 ± 11	T1	3 T	—
[31]	6	32 ± 13	FLAIR	3 T	
[32]	17	range from 17 to 53	T1	1.5 T–3 T	—
[33]	25	range from 17 to 59	T1, T2	3 T	—
[34]	70	range from 18 to 60	T1, T2	3 T	—
[35]	39	pediatric median 13 (IQR 13–14), adults median 37 (IQR 32–42)	T1	3 T	—
[36]	104	32.3 ± 14.2	T1, QPET	1.5 T–3 T	—
[37]	78	14.2 ± 4.5	T1	1.5 T–3 T	—
[38]	18	34 ± 2.5	T1	1.5 T	—
[39,40]	21	—	T1	—	—
[41]	11	range from 5 to 38	T1, FLAIR	3 T	—
[42]	54	range from 6.45 to 17.11	T1	3 T	—
[43]	7	33 ± 12	FLAIR	3 T	—
[44]	41	—	T1	1.5 T	—
[45]	11	—	T1	1.5 T	—
[46]	28	26.5 ± 14.1	T1	3 T	—
[47]	41	27 ± 9	T1, T2, FLAIR	3 T	—
[48]	22	14.68 ± 9.12	T1	3 T	available upon request
[49]	19	29 ± 8	T1, T2	1.5 T–3 T	—
[50,51]	10	—	T1	3 T	—
[52]	46	27.1 ± 8.6	T1, FLAIR	3 T	—
[53]	62	—	T1	3 T	—
[54]	43	24 ± 10	T1	1.5 T	—
[55,56]	41	24.9 ± 10.9	T1	1.5 T	—
[57]	22	12.1 ± 3.9	T1, FLAIR	1.5 T	—
[58]	40	—	T1, FLAIR, PET	3 T	—
[59]	34	range from 3.6 to 18.5	T1, FLAIR	3 T	—
[60]	113	29.5 ± 13.6	T1	1.5 T–3 T	available upon request
[61]	107	27 ± 9	T1, FLAIR	1.5 T–3 T	—
[62]	43	—	FLAIR	1.5 T–3 T	—
[63]	10	—	T1	1.5 T	—
[64]	19	24 ± 10	FLAIR	1.5 T–3 T	—
[65]	30	—	—	3 T	—
[66]	26	—	FLAIR	3 T	—
[67]	171	range from 2 to 55	T1, FLAIR	1.5 T–3 T	available upon request
[68]	201	range from 8 to 68	T1, FLAIR	1.5 T–3 T	—
[69]	80	11.5 ± 3.22	T1, T2, FLAIR	3 T	—

**Table 2 sensors-23-07072-t002:** Results of morphometric analyses using SPM obtained in different studies.

Study	Technique	Sensitivity %	Sequence	Patients	Controls	Age Group
[32]	SPM	53	T1	17	64	Adult patients
[33]	SPM-5	88	T2-FLAIR	25	25	Adult patients
[35]	SPM-12	92	T1	39	105	Adult patients/Pediatric patients
[36]	SPM-12	74	T1	104	N/A	Adult patients
[37]	SPM-12	56	T1	78	370	Pediatric patients

**Table 3 sensors-23-07072-t003:** Results of different Bayes classifier implementations.

Study	Technique	Sensitivity %	Patients	Age Group
[38]	Two-stage Bayes	85	18	Adult patients
[39]	Naïve Bayes	62.49	21	Adult patients
[40]	Naïve Bayes + SVM	88	21	Adult patients
[41]	Naïve Bayes	51	11	Pediatric patients/Adult patients
[42]	Two-stage Bayes	70	54	Pediatric patients
[43]	Three-stage Bayes	87.5	7	Adult patients

**Table 4 sensors-23-07072-t004:** Results of different SVM classifier implementations.

Study	Technique	Sensitivity %	Sequence	Patients	Age Group
[40]	Naïve Bayes + SVM	88	T1	21	Adult patients
[44]	SVM	98	T1	41	Adult patients
[45]	OC-SVM	77	T1	11 (13 lesions)	Adult patients
[46]	SVM-Mat	93	T1, PET	28	Adult patients

**Table 5 sensors-23-07072-t005:** Results of the different ensemble classifier implementations.

Study	Technique	Sensitivity %	Sequence	Patients
[47]	Two-stage RUSBoost	92	T1, T2, FLAIR	41
[48]	XGBoost	93	T1, FLAIR, PET	22

**Table 6 sensors-23-07072-t006:** Results of different CNN architectures.

Study	Detection Rate		Sensitivity %		Sequence	Patients	Controls	Age Group
	Training	Testing	Training	Testing				
[61]	35/40	61/67	87	91	T1, FLAIR	107	101	Pediatric patients/Adult patients
[63]	-	0.9	-	0.9	T1	10	20	—
[62]	33/40	-	82.5	-	FLAIR	43	-	—
[12]	-	-	100	92.45	T1	10	95	Pediatric patients/Adult patients
[64]	-	15/18	-	83.33	FLAIR	19	-	Pediatric patients/Adult patients
[65]	-	11/15	-	73.3	-	30	17	—
[66]	-	-	-	92	FLAIR	26	-	—
[67]	137/148	19/23	93	83	T1, FLAIR	171	131	Pediatric patients/Adult patients
[68]	-	-	70.1	77.8	T1, FLAIR	201	150	Pediatric patients/Adult patients
[13]	-	-	-	96.47	-	15	-	—
[69]	77/80	4/6	96	67	T1, T2, FLAIR	80	15	Pediatric patients

## Data Availability

Not applicable.

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
