# Peer review of "Automatic Detection of Focal Cortical Dysplasia Using MRI: A Systematic Review"

_sensors, 2023, doi:10.3390/s23167072_

Round 1

Reviewer 2 Report

This paper provides a comprehensive review of focal cortical dysplasia detection.

Some details of this review need to be clarified.
1. Line 80: Please elaborate on the difference between this view and the existing review [10].
2. In all tables, Do these methods use the same benchmark dataset for a fair comparison?

This review just "described" each paper one by one without insightful comments. In addition, this review lacks discussions in the following aspects:
1. What are the major benchmark datasets used in this research area?
2. What are performance metrics?
3. What is the future research direction?

NA

Round 2

Reviewer 1 Report

The authors incorporated all the points raised in the first round.

Reviewer 2 Report

The revision is good, there is no further comment.

NA